1    **Modeling groundwater responses to climate change in the Prairie Pothole Region**

Zhe Zhang[1], Yanping Li[1*], Michael Barlage[2], Fei Chen[2], Gonzalo Miguez-Macho[3], Andrew Ireson[1],
Zhenhua Li[1]
*[1]Global Institute for Water Security, University of Saskatchewan, Saskatoon, SK, Canada*
*[2]National Center for Atmospheric Research, Boulder, Colorado, USA*
*[3]Nonlinear Physic Group, Faculty of Physics, Universidade de Santiago de Compostela, Galicia, Spain*
Abstract
Shallow groundwater in the Prairie Pothole Region (PPR) is recharged predominantly by snowmelt
in the spring and supplies water for evapotranspiration through the summer and fall. This two-way
exchange is underrepresented in current land surface models. Furthermore, the impacts of climate
change on the groundwater recharge rates are uncertain. In this paper, we use a coupled land and
groundwater model to investigate the hydrological cycle of shallow groundwater in the PPR and
study its response to climate change at the end of the 21st century. The results show that the model
does a reasonably good job of simulating the timing of recharge. The mean water table depth
(WTD) is well simulated, except the model predicts deep WTD in northwestern Alberta. The most
significant change under future climate conditions occurs in the winter, when warmer temperature
changes the rain/snow partitioning,  delaying the time for snow accumulation/soil freezing while
advancing early melting/thawing. Such changes lead to an earlier start to a longer recharge season,
but with lower recharge rates. Different signals are shown in the eastern and western PPR in the
future summer, with reduced precipitation and drier soils in the east but little change in the west.
The annual recharge increased by 25% and 50% in the eastern and western PPR, respectively.
Additionally,  we found  the mean and seasonal variation of the simulated WTD are sensitive to
soil properties and fine-scale soil information is needed to improve groundwater simulation on
regional scale.
Keywords: Groundwater, Recharge, Climate Change, Prairie Pothole Region, Hydrological Cycle,
Introduction
The Prairie Pothole Region (PPR) in North America is located in a semi-arid and cold region,
where evapotranspiration (ET) exceeds precipitation (PR) in summer and near-surface soil is
frozen in winter (Gray, 1970; Granger and Gray, 1989; Hayashi et al., 2003; Pomeroy et al., 2007;
Ireson et al., 2013; Dumanski et al., 2015). These climatic conditions have introduced unique
hydrological characters to the groundwater flow in the PPR (Ireson et al., 2013). During winters,
frozen soils reduce permeability and snow accumulates on the surface, prohibiting infiltration (Niu
and Yang 2006; Mohammed et al., 2018). At the same time, the water table slowly declines due to
a combination of upward transport to the freezing front by the capillary effect and discharge to
rivers (Ireson et al., 2013). In early spring, snowmelt becomes the dominant component of the
hydrological cycle and the melt water runs over frozen soil, with little infiltration contributing to
recharge. As the soil thaws, the increased infiltration capacity allows snowmelt recharge to the
water table, the previously upward water movement by capillary effect to reverse and move
downwards, and the water table to rise to its maximum level. In summer and fall, when high ET
exceeds PR, capillary rise may draw water from the groundwater aquifers to supply ET demands,
declining water table. These processes characterize the critical two-way water exchange between
the unsaturated soils and saturated groundwater aquifers.

Previous studies have suggested that substantial changes to groundwater interactions with
unsaturated soils are likely to occur under climate change (Tremblay et al., 2011; Green et al.,
2011; Ireson et al., 2013, 2015). Existing modeling studies on the impacts of climate change on
groundwater are either at global or basin/location-specific scales (Meixner et al., 2016). Global-
level groundwater studies focus on potential future recharge trends (Doll and Fiedler, 2008; Doll,
2009; Green et al., 2011), yet coarse resolution analysis from global climate models (GCMs)
provided insufficient specificity to inform decision making. Basin-scale groundwater studies
connect the climate with groundwater-flow models to understand the climate impacts on specific
systems (Maxwell and Kollet, 2008; Kurylyk and MacQuarrie, 2013; Dumanski et al., 2015).
Regional groundwater modeling studies, such as in the Colorado River Basin (Christensen et al.,
2004) and in the western U.S. (Niraula et al., 2017), have applied downscaled climate scenarios
from GCMs to drive large scale hydrology models. These studies identified research gaps
associated with poor representation of groundwater-soil interactions in models and uncertainties
in future climate projections.

It is challenging to represent groundwater flows in LSMs because the important two-way water
exchange between unsaturated soils and groundwater aquifers was neglected in previous LSMs.
Recently, this two-way exchange has been implemented in coupled land surface – groundwater
models (LSM-GW). For example, Maxwell and Miller (2005) used a groundwater model (ParFlow)
coupled with the Common Land Model (CLM) as a single column model. They found that the
coupled and uncoupled models were very similar in simulated sensible heat flux (SH), ET, and
shallow soil moisture (SM), but differed greatly in simulated runoff and deep SM. Later on, Kollet
and Maxwell (2008) incorporated the ET effect on redistributing moisture upward from shallow
water table depth (WTD) and found the surface energy partitioning is highly sensitive to the WTD
when the WTD is less than 5 m below ground surface. Niu et al. (2011) implemented a simple
groundwater model (SIMGM, Niu et al., 2007), into the community Noah LSM with multi-
parameterization options (Noah-MP LSM), by adding an unconfined aquifer at the bottom of soil
layers. More complex features such as three-dimensional subsurface flow and two-dimensional
surface were included in ParFlow v3 and evaluated over much of continental North America for a
very fine 1-km resolution (Maxwell et al., 2015). These recent development in coupled land and
groundwater models have advanced our knowledge on the important interactions between soil and
groundwater aquifer.

In cold regions, soil freeze-thaw processes further complicate this two-way exchange. Field studies
have found that frozen soil not only influences the timing and amount of downward recharge to
aquifers by reducing the soil permeability (Koren et al., 1999; Niu et al., 2006; Kelln et al., 2007),
but may also induce upward water transport from aquifers to soil freezing fronts (Spaans and Baker,
1996; Remenda et al., 1996; Hansson et al., 2004). In the modeling community, a range of
approaches have been applied to deal with frozen soil parameterizations. Earlier LSMs assumed
no significant heat transfer and soil water redistribution for sub-freezing temperature, for example,
in simplified SiB and BATS (Xue et al., 1991; Dickinson et al., 1993; Niu and Zeng, 2012). Koren
et al. (1999) suggested that the frozen soil is permeable due to macropores that exist in soil
structural aggregates, such as cracks, dead root passages, and worm holes. The NoahV3 model
adopted this scheme as its default option. Niu and Yang (2006) suggested to separate a model grid
into frozen and unfrozen patches, and these two patches have a linear effect on the soil hydraulic
properties. This treatment was incorporated into CLM 3.0 and Noah-MP in 2007 and 2011,
respectively.

The spatial heterogeneity of soil moisture and WTD requires high-resolution meteorological input
that direct outputs from GCMs are too coarse to provide. In GCMs, differences in simulated
precipitation stem from the choice of convection parameterization scheme (Sherwood et al., 2014;
Prein et al., 2015). An important approach to improve precipitation simulation is to conduct
dynamical downscaling using the convection-permitting model (CPM) (Ban et al., 2014; Prein et
al., 2015; Liu et al., 2017). The CPM uses a high spatial resolution (usually under 5-km) to
explicitly resolve convection without activating convection parameterization schemes. CPMs can
also improve the representation of fine-scale topography and spatial variations of surface fields
(Prein et al., 2013). These CPM added-values provide an excellent opportunity to investigate water
table dynamics in the PPR.

The objectives of this paper are to 1) investigate the performance of a regional scale coupled land-
groundwater model in simulating groundwater  water levels, recharge and storage in a seasonally
frozen environment in PPR; and 2) explore the possible impacts of climate change on these
processes.

In this paper, we use a physical process-based LSM (Noah-MP) coupled with a groundwater
dynamics model (MMF model). The coupled Noah-MP-MMF model is driven by two sets of
meteorological forcing for 13 years under current and future climate scenarios. These two sets of
meteorological dataset are from a CPM dynamical downscaling project using the Weather
Research & Forecast (WRF) model with 4-km grid spacing covering the Contiguous U.S. and
Southern Canada (WRF CONUS, Liu et al., 2017). The paper is structured as follows: Section 2
introduces the groundwater observations for WTD evaluation in the PPR, the coupled Noah-MP-
MMF model, and the meteorological forcing from the WRF CONUS project. Section 3 evaluates
the model simulated WTD timeseries and shows the groundwater budget and hydrological changes
due to climate change. Section 4 and 5 offer a broad discussion and conclusion.
2. Data and Methods
2.1 Observational data
Groundwater observation data were obtained through several agencies: (1) the United States
Geological Survey (USGS) National Water Information System in the U.S.
(https://waterdata.usgs.gov/nwis/gw),          (2)          the          Alberta          Environment
(http://aep.alberta.ca/water/programs-and-services/groundwater/groundwater-observation-well-
network/default.aspx),          (3)          the          Saskatchewan          Water          Security          Agency
(https://www.wsask.ca/Water-Info/Ground-Water/Observation-Wells/).

Initially, groundwater data from 160 wells were acquired, 72 in the U.S., 43 from Alberta, and 45
from Saskatchewan. We used the following criteria to select qualified stations for our study and
evaluate our model performance against these observations:

1)  the location of the wells are within the PPR region;

2)  a sufficiently long data record exists during the simulation period. We define the

observation availability as the available observation period within the 13-year simulation

period and select wells with observation availability greater than 80%;

3)  we only take data from unconfined aquifers with shallow groundwater levels (mean WTD >

5 m);

4)  we only take data with minimal anthropogenic effects (such as from pumping or irrigation).

These criteria reduced the observation data to 33 well records, with six in Alberta, 13 in
Saskatchewan and 14 from the U.S. **Table 1** summarizes the information for each selected well,
and **Fig. 1**(a) shows the location of the wells in our study area. It is noteworthy that most of the
groundwater sites have more permeable deposits (sand and gravel) as provincial and state agencies
don't monitor low permeability formations. More information about the selecting criteria are
provided in the supplemental materials.

**Fig. 1** (a) Topography of the Prairie Pothole Region (PPR) and station location of rain gauges (black dots) and
groundwater wells (red diamonds); (b) Topography of the WRF CONUS domain, with the black box indicating the
PPR domain.

**Table 1.** Summary of the locations and aquifer type and soil type of the 33 selected wells.

2.2 Groundwater and Frozen Soil Scheme in Noah-MP
In the present study, we used the community Noah-MP LSM (Niu et al. 2011; Yang et al. 2011),
coupled with a GW model – the MMF model (Fan et al. 2007; Miguez-Macho et al., 2007). This
coupled model has been applied in many regional hydrology studies in offline mode (Miguez-
Macho and Fan 2012; Martinez et al., 2016) and coupled with regional climate models (Anyah et
al., 2008; Barlage et al., 2015). We present here a brief introduction to the MMF groundwater
scheme and the frozen soil scheme in Noah-MP, further details can be found in previous studies
(Fan et al., 2007;Miguez-Macho et al., 2007; Niu and Yang, 2006).

Fig. 2 is a diagram of the structure of 4 soil layers (0.1, 0.3, 0.6 and 1.0 m) and the underlying
unconfined aquifer in Noah-MP-MMF. The MMF scheme defines explicitly an unconfined aquifer
below the 2-m soil and an auxiliary soil layer stretching to the WTD, which varies in space and
time [m]. The thickness of this auxiliary layer ($z_{aux}$ [m]) is also variable, depending on the WTD:
$$z_{aux} = \begin{cases} 1, & WTD \geq -3 \\ -2 - WTD, & WTD < -3 \end{cases} \quad (1)$$

The vertical fluxes include gravity drainage and capillary flux, solved from the Richards' equation,
$$q = K_\theta \left( \frac{\partial \psi}{\partial z} - 1 \right), \quad K_\theta = K_{sat} * \left( \frac{\theta}{\theta_{sat}} \right)^{2b+3}, \quad \psi = \psi_{sat} * \left( \frac{\theta_{sat}}{\theta} \right)^b \quad (2)$$
where $q$ is water flux between two adjacent layers [m/s], $K_\theta$ is the hydraulic conductivity [m/s] at
certain soil moisture content $\theta$ [m³/m³], $\psi$ is the soil matric potential [m] and $b$ is soil pore size
index. The subscript *sat* denotes saturation. The recharge flux from/to the layer above WTD, $R$,
can be obtained according to WTD:
$$R = \begin{cases} K_k * \left( \dfrac{\psi_i - \psi_k}{z_{soil(i)} - z_{soil(k)}} - 1 \right), & WTD \geq -2 \\[2mm] K_{aux} * \left( \dfrac{\psi_4 - \psi_{aux}}{(-2) - (-3)} - 1 \right), & -2 > WTD \geq -3 \\[2mm] K_{sat} * \left( \dfrac{\psi_{aux} - \psi_{sat}}{(-2) - (WTD)} - 1 \right), & WTD < -3 \end{cases} \quad (3)$$


In the first case, WTD is in the resolved soil layers and $z_{soil}$ is the depth of soil layer with the
subscript $k$ indicating the layer containing WTD while $i$ is the layer above. The calculated water
table recharge is then passed to the MMF groundwater routine.

The change of groundwater storage in the unconfined aquifer considers three components:
recharge flux ($R$), river discharge ($Q_r$), and lateral flows ($Q_{lat}$):
$$\Delta S_g = (R - Q_r + \sum Q_{lat}) \quad (4)$$

where $S_g$ [mm] is groundwater storage, $Q_r$ [mm] is the water flux of groundwater-river exchange,
and $\sum Q_{lat}$ [mm] are groundwater lateral flows to/from all surrounding grid cells. The groundwater
lateral flow ($\sum Q_{lat}$) is the total horizontal flows between each grid cell and its neighbouring grid
cells, calculated from Darcy's law with the Dupuit–Forchheimer approximation (Fan and Miguez-
Macho 2010), as:
$$Q_{lat} = wT \left( \frac{h - h_n}{l} \right) \quad (5)$$

where $w$ is the width of cell interface [m], $T$ is the transmissivity of groundwater flow [m$^2$/s], $h$
and $h_n$ are the water table head [m] of local and neighboring cell, and $l$ is the length [m] between
cells. $T$ depends on hydraulic conductivity $K$ and WTD:

$$T = \begin{cases} \int_{-\infty}^{h} K \, dz & WTD \geq -2 \\ \int_{-\infty}^{(z_{surf}-2)} K \, dz + \sum K_i * dz_i & WTD < -2 \end{cases} \quad (7)$$

For $WTD < -2$, $K$ is assumed to decay exponentially with depth, $K = K_4 \exp(-z/f)$, $K_4$ is the

hydraulic conductivity in the 4-th soil layer and $f$ is the e-folding length and depends on terrain

slope. For WTD ≥ -2, $i$ represents the number of layers between the water table and the 2-m bottom

and $z_{surf}$ is the surface elevation.

The river flux ($Q_r$) is also represented by a Darcy's law–type equation, where the flux depends on

the gradient between the groundwater and the river depth and the riverbed conductance:

$$Q_r = RC \cdot (h - z_{river}) \quad (8)$$

with $z_{river}$ is the depth of river [m] and $RC$ is dimensionless river conductance, which depends on

the slope of the terrain and equilibrium water table. Eq. (8) is a simplification which uses $z_{river}$

rather than the water level in the river and, for this study, we only consider one-way discharge

from groundwater to rivers. Finally, the change of WTD is calculated as the total fluxes fill or

drain the pore space between saturation and the equilibrium soil moisture state ($\theta_{eq}$ [m³/m³]) in

the layer containing WTD:

$$\Delta \text{WTD} = \frac{\Delta S_g}{(\theta_{sat} - \theta_{eq})} \quad (9)$$

If $\Delta S_g$ is greater than the pore space in the current layer, the soil moisture content of current layer

is saturated and the WTD rises to the layer above, updating the soil moisture content in the layer

above as well. Vice versa for negative $\Delta S_g$ as water table declines and soil moisture decreases.

**Fig. 2** Structure of the Noah-MP LSM coupled with MMF groundwater scheme, the top 2-m soil of 4 layers whose
thicknesses are 0.1, 0.3, 0.6 and 1.0 m. An unconfined aquifer is added below the 2-m boundary, including an auxiliary
layer and the saturated aquifer. Positive flux of $R$ denotes downward transport. Two water table are shown, one within
the 2-m soil and one below, indicating that the model is capable to deal with both shallow and deep water table.

There are two options in Noah-MP LSM for frozen soil permeability; option 1, the default option
in Noah-MP, is from Niu and Yang (2006) and option 2 is inherited the Koren et al. (1999) scheme
from NoahV3. Option 1 assumes that a model grid cell consists of permeable and impermeable
patches and the area weighted sum of these patches gives the grid cell soil hydraulic properties.
Thus, the total soil moisture ($\theta$) in the grid cell is used to compute hydraulic properties as:
$$\theta = \theta_{ice} + \theta_{liq} \qquad (10)$$

$$K = \left(1 - F_{frz}\right)K_u = \left(1 - F_{frz}\right)K_{sat}\left(\frac{\theta}{\theta_{sat}}\right)^{2b+3} \quad (11)$$

the subscript $frz$ and $u$ denote the frozen and unfrozen patches in the grid point. The impermeable
frozen soil fraction is parameterized as:
$$F_{frz} = e^{-\alpha(1-\theta_{ice}/\theta_{sat})} - e^{-\alpha} \qquad (12)$$

$\alpha = 3.0$ is an adjustable parameter. The amount of the liquid water in soil layer is either $\theta_{liq}$ or
$\theta_{liq,max}$, the maximum amount of liquid water, which is calculated by a more general form of the
freezing-point depression equation:
$$\theta_{liq,max} = \theta_{sat}\left\{\frac{10^3 L_f\left(T_{soil} - T_{frz}\right)}{gT_{soil}\psi_{sat}}\right\}^{-\frac{1}{b}} \qquad (13)$$

where $T_{soil}$ and $T_{frz}$ are soil temperature and freezing point [K]; $L_f$ is the latent heat of fusion [J
kg$^{-1}$]; g is gravitational acceleration [m s$^{-2}$].

On the other hand, the option 2 uses only the liquid water volume to calculate hydraulic properties
and assumes a non-linear effect of frozen soil on permeability. Also, the option 2 uses a variant of
freezing-point depression equation with an extra term, $(1 + 8\theta_{ice})^2$, to account for the increased
interface between soil particles and liquid water due to the increase of ice crystals. Generally,
option 1 assumes that soil ice has a smaller effect on infiltration and simulates more permeable
frozen soil than option 2 (Niu et al., 2011). For this reason, the option 1 allows the soil water to
move and redistribute more easily within the frozen soil and we decide to use option 1 in our study.
2.3 Forcing Data
The output from the WRF CONUS dataset (Liu et al. 2017) are used as meteorological forcing to
drive the Noah-MP-MMF model. The WRF CONUS project consists of two simulations. The first
simulation is referred as the current climate scenario, or control run (CTRL), from Oct 2000 to Sep
2013, and forced with the 6-hourly 0.7° ERA-Interim reanalysis data. The second simulation is a
perturbation to reflect the future climate scenario, closely following the pseudo global warming
(PGW) approach in previous works (Rasmussen et al., 2014). The PGW simulation is forced with
6-hourly ERA-Interim reanalysis data plus a delta climate change signal derived from an ensemble
of CMIP5 models under the RCP8.5 emission scenario and reflects the climate change signal
between the end of $21^{st}$ and $20^{th}$ century.

**Fig. 3** shows the annual precipitation in the PPR from 4-km WRF CONUS from the current climate
and 32-km North America Regional Reanalysis (NARR, another reanalysis dataset commonly
used for land surface model forcing). Both datasets show similar annual precipitation pattern and
bias patterns compared to observations: underestimating of precipitation in the east and
overestimating in the west. However, the WRF CONUS shows significant improvement of
percentage bias in precipitation ((Model-Observation)/Observation) over the western PPR. For the
consistency of the same source of data for current and future climate, the WRF-CONUS is the best
available dataset for the coupled land-groundwater study in the PPR.

**Fig. 3** Evaluation of the annual precipitation from WRF CONUS (top) and NARR (bottom) against rain gauge
observation.

For the future climate study, the precipitation and temperature of the PGW climate forcing are
shown in **Fig. 4** and **Fig. 5**. The WRF CONUS projects more precipitation in the PPR, except in
the southeast of the domain in summer, where it shows a precipitation reduction of about 50 to 100
mm. On the other hand, the WRF CONUS projects strongest warming occurring in the northeast
PPR in winter (**Fig. 5**), about 6–8 °C. Another significant warming signal occurs in summer in the
southeast of domain, corresponding to the reduction of future precipitation, as seen in **Fig. 4**.

**Fig. 4** Seasonal accumulated precipitation from current climate scenario(CTRL), future climate scenario (PGW) and
projected change (PGW-CTRL) in the forcing data.

**Fig. 5** Seasonal averaged temperature from CTRL, PGW, and the projected change (PGW-CTRL).

2.4 Model Setup
The two Noah-MP-MMF simulations representing the current climate and future climate are
denoted as CTRL and PGW, respectively. The initial groundwater levels are from a global 1-km
equilibrium groundwater map (Fan et al., 2013) and the equilibrium soil moisture for each soil
layer is calculated at the first model timestep with climatology recharge, spinning up for 500 years.
Since the model domain is at a different resolution than the input data, the appropriate initial WTD
at 4-km may be different than the average at 1-km. To properly initialize the simulation, we spin
up the model using the forcing of current climate (CTRL) for the years from 2000 to 2001
repeatedly (in total 10 loops).

Due to different data sources, the default soil types along the boundary between the U.S. and
Canada are discontinuous. Thus, we use the global 1-km fine soil data (Shangguan et al., 2014,
http://globalchange.bnu.edu.cn/research/soilw) in our study region. The soil properties for the
aquifer use the same properties as the lowest soil layer from the Noah-MP 2-m soil layers.
3. Results
3.1 Comparison with groundwater observations
According to the locations of 33 groundwater wells in **Table 1**, the simulated WTD from the
closest model grid points are extracted. **Fig. 6** shows the modeled WTD bias from the CTRL run.
We also select the monthly WTD timeseries from 8 sites, the observation are in black dots and
CTRL in blue lines. See supplemental materials for the timeseries of 33 sites. The model produces
reasonable values of mean WTD, the mean bias are smaller than 1 m in most of sites, except in
Alberta, where the model predicts deep bias about 5 m in the northwestern part of PPR. The model
also successfully captures the annual cycle of WTD, which rises in spring and early summer,
because of snowmelt and rainfall recharge, and declines in summer and fall, because of high ET,
and in winter because of frozen near-surface soil. In all observations, the timing of the water table
rising and dropping is well simulated, as the timing and amount of infiltration and recharge in
spring is controlled by the freeze-thaw processes in seasonally frozen soil.

**Fig. 6.** WTD (m) bias from CTRL simulation and timeseries from 8 groundwater wells in PPR (black for observation and
blue for CTRL model simulation). See Table 2 CTRL column for the model statistics and supplemental materials for complete
timeseries from 33 wells.

On the other hand, the model simulated WTD seasonal variation is smaller than observations. The
small seasonal variation could be due to the misrepresentation between the lithology from the
observational surveys and the soil types in the model grids. As mentioned in Section 2.2, the
groundwater aquifer uses the same soil types as the bottom layer of the resolved 2-m soil layers.
While sand and gravel are the dominant lithology in most of the sites, they are mostly clay and
loam in the model (Table 1). For sandy soil reported in most of the sites, small capacity and fast
responses to infiltration lead to large water table fluctuations, whereas, in the model, clay and loam
soil allows low permeability and large capacity, and smoothens responses to recharge and capillary
effects. Furthermore, the 4-layer soils are vertically homogeneous in soil type and the groundwater
model uses the lowest level soil type as the aquifer lithology. For many part of the PPR, where
groundwater level are perched at the top 5-m due to a layer called glacial till. These
geohydrological characteristics cannot be reflected in this model and contribute to the deep WTD
bias simulated in Alberta. This shortcoming of the model was also reported in a study taken place
in the Amazon rainforest (Miguez-Macho et al., 2012).

3.2 Climate change signal in Groundwater fluxes
The MMF groundwater model simulates three components in the groundwater water budget, the
recharge flux ($R$), lateral flow ($Q_{lat}$), and discharge flux to rivers ($Q_r$). Because the topography is
usually flat in the PPR, the magnitude of groundwater lateral transport is very small ($Q_{lat}$ less than
5 mm per year). On the other hand, the shallow water table in the PPR region is higher than the
local river bed, thus, the $Q_r$ term is always discharging from groundwater aquifers to rivers. As a
result, the recharge term is the major contributor to the groundwater storage in the PPR, and its
variation (usually between -100 to 100 mm) dominates the timing and amplitude of the water table
dynamics. The seasonal accumulated total groundwater fluxes in the  PPR ($R+Q_{lat} - Q_r$) are
shown in **Fig. 7**. The positive (negative) flux in blue (red) means the groundwater aquifer is gaining
(losing) water, causing the water table to rise (decline).

**Fig. 7** Seasonal accumulated total groundwater fluxes ($R+$ ) for current climate (CTRL, top), future climate (PGW,
middle) and projected change (PGW-CTRL, bottom) in forcing data. Black dashed lines in PGW-CTRL separate the
PPR into eastern and western halves.

Under current climate conditions, the total groundwater fluxes show strong seasonal fluctuations,
consistent with the WTD timeseries shown in **Fig. 6**. On average, in fall (SON) and winter (DJF),
there is a 20-mm negative recharge, driven by the capillary effects and drawing water from aquifer
to dry soil above. Spring (MAM) is usually the season with a strong positive recharge because
snowmelt provides a significant amount of water, and soils thawing allow infiltration. The large
amount of snowmelt water contributes to more than 100 mm of positive recharge in the eastern
domain. It is until summer (JJA), when strong ET depletes soil moisture and results in about 50
mm of negative recharge.

Under future climate conditions, the increased PR in fall and winter leads to wetter upper soil
layers, resulting in a net positive recharge flux (PGW – CTRL in SON and DJF). However, the
PGW summer is impacted by increased ET under a warmer and drier climate, due to higher
temperature and less PR. As a result, the groundwater uptake by the capillary effect is more critical
in the future summer. Furthermore, there is a strong east-to-west difference in the total
groundwater flux change from PGW to CTRL. In the eastern PPR, the change in total groundwater
flux exhibits obvious seasonality while the model projects persistent positive groundwater fluxes
in the western PPR.

3.3 Water budget analysis
**Fig. 8** and **Fig. 9** show the water budget analysis for the eastern and western PPR (divided by the
dotted line in 103° W in Fig. 7), respectively. Four components are presented in the figures, i.e.
(1) PR and ET; (2) surface and underground runoff (*SFCRUN* and *UDGRUN*); and surface
snowpack; (3) the change of soil moisture storage and (4) groundwater fluxes and the change of
storage. In the current and future climate, these budget terms are plotted in annual accumulation
((a) and (b) for CTRL and PGW), whereas their difference are plotted in each month individually
((c) for PGW-CTRL).

Under current climate conditions, during snowmelt infiltration and rainfall events, water infiltrates
into the top soil layer, travels through the soil column and exits the bottom of the 2-m boundary,
hence, the water table rises. During the summer dry season, ET is higher than PR and the soil
layers lose water through ET, therefore, the capillary effect takes water from the underlying aquifer
and the water table declines. In winter, the near-surface soil in the PPR is seasonally frozen, thus,
a redistribution of subsurface water to the freezing front results in negative recharge, and the water
table declines.

In the eastern PPR, the effective precipitation (PR-ET) is found to increase from fall to spring, but
decrease in summer in PGW (**Fig. 8**(1c)). Warmer falls and winters in PGW, together with
increased PR, not only delay snow accumulation and bring forward snowmelt,  but also change
the precipitation partition – more as rain and less as snow. This warming causes up to 20 mm of
snowpack loss (**Fig.8**(2c)). The underground runoff starts much earlier in PGW (December)
(**Fig.8**(2b)) than in CTRL (February) (**Fig.8**(2a)). On the other hand, the warming in PGW also
changes the partitioning of soil ice and soil water in unsaturated soil layers (**Fig**. 8(3c)). For late
spring in PGW, the springtime recharge in the future is significantly reduced due to early melting
and less snowpack remaining (**Fig. 8**(4c)). In the PGW summer, reduced PR (50 mm less) and
higher temperatures (8 °C warmer) lead to reduction in total soil moisture, and a stronger negative
recharge from the aquifer. Therefore, the increase of recharge from fall to early spring compensates
the recharge reduction due to stronger ET in summer in the eastern PPR, and changes little in the
annual mean groundwater storage (1.763 mm per year).

**Fig. 8** Water budget analysis in the eastern PPR in (a) CTRL, (b) PGW and (c) PGW – CTRL. Water budget terms
include: (1) *PR & ET*, (2) surface snow, surface runoff and underground runoff (*SNOW*, *SFCRUN*, and *UDGRUN*),
(3) change of soil moisture storage (soil water, soil ice and total soil moisture, $\Delta SMC$) and (4) groundwater fluxes
and the change of groundwater storage ($R, Q_{lat}, Q_r, \Delta S_g$). The annual mean soil moisture change (PGW-CTRL) is
shown with black dashed line in (3). The Residual term is defined as $Res = (R+Q_{lat}-Q_r)-\Delta S_g$ in (4). Note that in (a)
and (b) the accumulated fluxes and change in storage are shown in lines, whereas in (c) the difference in (PGW-CTRL)
is shown for each individual month in bars.

These changes in water budget components in the western PPR (**Fig. 9**) are similar to those in the
eastern PPR (**Fig. 8**), except in summer. The reduction in summer PR in the western the PPR (less
than 5 mm reduction) is not as obvious as that in the eastern PPR (50 mm reduction) (**Fig. 4**). Thus,
annual mean total soil moisture in future is about the same as in current climate (Fig. 9(3c)) and
results in little negative recharge in PGW summer (**Fig. 9**(4c)). Therefore, the increase in annual
recharge is more significant (10 mm per year), an increase of about 50% of the annual recharge in
the current climate (20 mm per year) (**Fig. 9**(4c)).

**Fig. 9** Same as Fig. 8. Water budget analysis in the **western PPR**: in (a) CTRL, (b) PGW and (c) PGW – CTRL.
Water budget terms include: (1) *PR & ET*, (2) surface snow, surface runoff and underground runoff (*SNOW*, *SFCRUN*,
and *UDGRUN*), (3) change of soil moisture storage (soil water, soil ice and total soil moisture, $\Delta SMC$) and (4)
groundwater fluxes and the change of groundwater storage ($R$, $Q_{lat}$, $Q_r$, $\Delta S_g$). The annual mean soil moisture change
(PGW-CTRL) is shown with black dashed line in (3). The Residual term is defined as $Res = (R+Q_{lat}-Q_r)-\Delta S_g$ in (4).
Note that in (a) and (b) the accumulated fluxes and change in storage are shown in lines, whereas in (c) the difference
in (PGW-CTRL) is shown for each individual month in bars.

In both the eastern and western PPR, the water budget components for the groundwater aquifer are
plotted in **Fig. 8**(4) and **Fig. 9** (4), with the changes of each flux (PGW-CTRL) printed at the
bottom. The groundwater lateral flow is a small term in areal average and has little impact on the
groundwater storage. Nearly half of the increased recharge in both the eastern and western PPR is
discharged to river flux ($Q_r$ = 2.26 mm out of $R$ = 4.15 mm in the eastern PPR and $Q_r$ = 5.20 mm
out of $R$ = 10.72 mm in western PPR). Therefore, the groundwater storage change in the eastern
PPR (1.76 mm per year) is not as great as that in the western PPR (5.39 mm per year).

These two regions of the PPR show differences in hydrological response to future climate because
of the spatial variation of the summer PR. As shown in both **Fig. 4** (PGW-CTRL), **Fig. 8**(1) and
**Fig. 9**(1), the reduction of future PR in summer in the eastern PPR is significant (50 mm). The
spatial difference of precipitation changes in the PPR further results in the recharge increase
doubling in the western PPR compared to the eastern PPR.

4. Discussion
4.1 Improving WTD Simulation
In Section 3.1, we show that the model is capable of simulating the mean WTD in most sites, yet
predicts deep groundwater in Alberta and underestimates its seasonal variation. These results may
be due to misrepresentations between model default soil type and the soil properties in the
observational wells. To test this theory, an additional simulation, REP, is conducted by replacing
the default soil types in the locations of these 33 groundwater wells with sand-type soil, which is
the dominant soil types reported from observational surveys. The timeseries of the REP and default
CTRL are shown in Fig. 10 (also see supplemental materials for the complete 33 sites) and a
summary of the mean and standard deviation of the two simulations are provided in Table 2.

**Fig. 10** Same as Fig. 6, WTD (m) bias from CTRL simulation and timeseries from 8 groundwater wells in PPR (black for
observation and blue for CTRL model simulation, and red for the replacing soil type simulation). REP is the additional
simulation by replacing the default soil type in the model with sandy soil type.

The REP simulation with sandy soil show two sensitive signals: (1) REP WTD are shallower than
the default simulation; (2) and exhibit stronger seasonal variation. These two signals can be
explained by the WTD equation in the MMF scheme:
$$\Delta WTD = \frac{\Delta(R + Q_{lat} - Q_r)}{(\theta_{sat} - \theta_{eq})} \quad (14)$$
Eq. (14) represents that the change of WTD in a period of time is calculated by the total
groundwater fluxes, $\Delta(R + Q_{lat} - Q_r)$, divided by the available soil moisture capacity of current
layer ($\theta_{sat} - \theta_{eq}$). In REP simulation, the parameters $\theta_{sat}$ for the dominant soil type in
observational sites (sand/gravel) is smaller than those in default model grids (clay loam, sandy
loam, loam, loamy sand, etc.). Therefore, changing the $\theta_{sat}$ is essentially reducing the storage in
the aquifer and soil in this model grid. Given the same amount of groundwater flux, in the REP
simulation, the mean WTD is higher and the seasonal variation is stronger than the default CTRL
run.

In the REP simulation, we replaced soil type only at a limited number of sites because the
geological survey data in high resolution and large area extent is not yet available for the whole
PPR. At point scale, the WTD responses to climate change over these limited number of sites show
diverse results and uncertainties (see supplemental materials). For the rest of the domain, the
default soil type from global 1-km soil map is used. The REP modifications of soil types at point-
scale have  small contribution to the water balance analysis (Fig. 8 & 9) at regional-scale. Our
results and conclusions for groundwater response to PGW doesn't change. We are currently
undertaking a soil property survey project in the PPR region to obtain soil properties at high spatial
resolution, both horizontal and vertical. This may provide better opportunity to improve WTD
simulation as well as assess climate-groundwater interaction in future studies.

4.2 Climate Change Impacts on Groundwater Hydrological Regime
The warming and increased precipitation in cold seasons in future climate lead to later snow
accumulation, higher recharge in winter and earlier melting in spring compared to current climate.
Such changes in snowpack loss have been hypothesized in mountainous as well as high-latitude
regions (Taylor et al 2013; Ireson et al., 2015; Meixner et al., 2016; Musselman et al., 2017). In
addition to the amount of recharge, the shift of recharge season is also noteworthy. Under current
climate conditions in spring, soil thawing (in March) is generally later than snowmelt (in February)
by a month in the PPR. Thus, the snowmelt water in pre-thaw spring would either re-freeze after
infiltrating into partially frozen soil or become surface runoff. Under the PGW climate, the warmer
winter and spring allows snowmelt and soil thaw to occur earlier in the middle of winter (in January
and February, respectively). As a result, the recharge season starts earlier in December, and last
longer until June, results in longer recharge season but with lower recharge rate.

Future projected increasing evapotranspiration demand in summer desiccates soil moisture,
resulting in more water uptake from aquifers to subsidize dry soil in the future summer. This
groundwater transport to soil moisture is similar to the "buffer effect" documented in an offline
study in the Amazon rainforest (Pokhrel et al., 2014). In , shallow water tables exist in the critical
zone, where WTD ranges from 1 to 5 meters below surface and could exert strong influence on
land energy and moisture fluxes feedback to the atmosphere (Kollet and Maxwell, 2008; Fan ,
2015). Previous coupled atmosphere-land-groundwater studies at 30-km resolution showed that
groundwater could support soil moisture during summer dry period, but has little impacts on
precipitation in Central U.S. (Barlage et al., 2015). It would be an interesting topic to study the
integrated impacts of shallow groundwater to regional climate in the convection permitting
resolution (resolution < 5-km).

4.3 Fine-scale interaction between groundwater and Prairie pothole wetlands
Furthermore, groundwater exchange with prairie pothole wetlands are complicated and critical in
the PPR. Numerous wetlands known as potholes or sloughs provide important ecosystem services,
such as providing wildlife habitats and groundwater recharge (Johnson et al., 2010). Shallow
groundwater aquifers may receive water from or lose water to prairie wetlands depending on the
hydrological setting. Depression-focused recharge generated by runoff from upland to depression
contributes to sufficient amount of water input to shallow groundwater (5-40 mm/year) (Hayashi
et al., 2016).

On the other hand, groundwater lateral flow exchange center of a wetland pond to its moist margin
is also an important component in the wetland water balance (van der Kamp and Hayashi, 2009;
Brannen, et al., 2015; Hayashi et al., 2016). However, this groundwater-wetland exchange
typically occurs on local scale (from 10 to 100 m) and thus, is challenging to represent in current
land surface models or climate models (resolution from 1 km to 100 km). In this paper, we focus
on the groundwater dynamics on regional scale, which is still unable to capture these small wetland
features in this study. We admit this limitation and are currently developing a sub-grid scheme to
represent small scale open water wetlands as a fraction within a grid cell and calculate its feedback
to regional environments. Future studies on this topic will provide valuable insights on these key
ecosystems and their interaction under climate change.

Conclusion
In this study, a coupled land-groundwater model is applied to simulate the interaction between the
groundwater aquifer and soil moisture in the PPR. The climate forcing is from a dynamical
downscaling project (WRF CONUS), which uses the convection-permitting model (CPM)
configuration in high resolution. The goal of this study is to investigate the groundwater responses
to climate change, and to identify the major processes that contribute to these responses in the PPR.
To our knowledge, this is the first study applying CPM forcing in a hydrology study in this region.
We have three main findings:

(1) the coupled land-groundwater model shows reliable simulation of mean WTD, however
underestimates the seasonal variation of the water table against well observations. This could be
attributed to several reasons, including misrepresentation of topography and soil types, as well as
vertical homogenous soil layers used in the model. We further conducted an additional simulation
(REP) by replacing the model default soil types with sand-type soil and the simulated WTDs were
improved in both mean and seasonal variation. However, inadequacy of soil properties in deeper
layer and higher spatial resolution is still a limitation.

(2) Recharge markedly increases due to projected increased PR, particularly from fall to spring
under future climate conditions. Strong east-west spatial variation exists in the annual recharge
increases, 25% in the eastern and 50% in the western PPR. This is due to the significant projected
PR reduction in PGW summer in the eastern PPR but little change in the western PPR. This PR
reduction leads to stronger ET demand, which draws more groundwater uptake due to the capillary
effect, results in negative recharge in the summer. Therefore, the increased recharge from fall to
spring is consumed by ET in summer, and results in little change in groundwater in the eastern
PPR, while gaining water in the western PPR.

(3) The timing of infiltration and recharge are critically impacted by the changes in freeze-thaw
processes. Increased precipitation, combined with higher winter temperatures, results in later snow
accumulation/soil freezing, partitioned more as rain than snow, and earlier snowmelt/soil thaw.
This leads to substantial loss of snowpack, shorter frozen soil season, and higher permeability in
soil allowing infiltration. Late accumulation/freezing and early melting/thawing leads to an early
start of a longer recharge season from December to June, but with a lower recharge rate.

Our study has some limitations where future studies are encouraged:
(1) Despite the large number of groundwater wells in PPR, only a few are suitable for long-term
evaluation, due to data quality, anthropogenic pumping, and length of data record. As remote
sensing techniques advance, observing terrestrial water storage anomalies derived from the
GRACE satellite may provide substantial information on WTD, although the GRACE information
needs to be downscaled to a finer scale before comparisons can be made with regional hydrology
models at km-scale (Pokhrel et al., 2013).

(2) This study is an offline study of climate change impacts on groundwater. It is important to
investigate how shallow groundwater in the earth's critical zone could interact with surface water
and energy exchange to the atmosphere and affect regional climate. This investigation would be
important to the central North America region (one of the land atmosphere coupling "hot spots",
Koster et al., 2004 ).

**Acknowledgments**
The authors Zhe Zhang, Yanping Li, Zhenhua Li gratefully acknowledge the support from the
Changing Cold Regions Network (CCRN) funded by the Natural Science and Engineering
Research Council of Canada (NSERC), as well as the Global Water Future project and Global
Institute of Water Security at University of Saskatchewan. Yanping Li acknowledge the support
from NSERC Discovery Grant. Fei Chen, Michael Barlage appreciate the support from the Water
System Program at the National Center for Atmospheric Research (NCAR), USDA NIFA Grants
2015-67003-23508 and 2015-67003-23460, NSF INFEW/T2 Grant #1739705, and NOAA CFDA
Grant #NA18OAR4590381. NCAR is sponsored by the National Science Foundation. Any
opinions, findings, conclusions or recommendations expressed in this publication are those of the
authors and do not necessarily reflect the views of the National Science Foundation.

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

Table and Figure
**Table 1.** Summary of the locations and aquifer type and soil type of the 33 selected wells.

| Site Name/ Site No. | Lat | Lon | Elev | Aquifer type | Aquifer Lithology | Model Elevation | Model Soil type |
|---|---|---|---|---|---|---|---|
| Devon 0162 | 53.41 | -113.76 | 700.0 | Unconfined | Sand | 697.366 | Sandy loam |
| Hardisty 0143 | 52.67 | -111.31 | 622.0 | Unconfined | Gravel | 633.079 | Loam |
| Kirkpatrick Lake 0229 | 51.95 | -111.44 | 744.5 | Semi-confined | Sandstone | 778.311 | Sandy loam |
| Metiskow 0267 | 52.42 | -110.60 | 677.5 | Unconfined | Sand | 679.516 | Loamy sand |
| Wagner 0172 | 53.56 | -113.82 | 670.0 | Surficial | Sand | 670.845 | Silt loam |
| Narrow Lake 252 | 54.60 | -113.63 | 640.0 | Unconfined | Sand | 701.0 | Clay loam |
| Baildon 060 | 50.25 | -105.50 | 590.184 | Surficial | - | 580.890 | Sandy loam |
| Beauval | 55.11 | -107.74 | 434.3 | Intertill | Sand | 446.5 | Sandy loam |
| Blucher | 52.03 | -106.20 | 521.061 | Intertill | Sand/Gravel | 523.217 | Loam |
| Crater Lake | 50.95 | -102.46 | 524.158 | Intertill | Sand/Gravel/Clay | 522.767 | Loam |
| Duck Lake | 52.92 | -106.23 | 502.920 | Surficial | Sand | 501.729 | Loamy sand |
| Forget | 49.70 | -102.85 | 606.552 | Surficial | Sand | 605.915 | Sandy loam |
| Garden Head | 49.74 | -108.52 | 899.160 | Bedrock | Sand/Till | 894.357 | Clay loam |
| Nokomis | 51.51 | -105.06 | 516.267 | Bedrock | Sand | 511.767 | Clay loam |
| Shaunavon | 49.69 | -108.50 | 896.040 | Bedrock | Sand/Till | 900.433 | Clay loam |
| Simpson 13 | 51.45 | -105.18 | 496.620 | Surficial | Sand | 493.313 | Sandy loam |
| Simpson 14 | 51.457 | -105.19 | 496.600 | Surficial | Sand | 493.313 | Sandy loam |
| Yorkton 517 | 51.17 | -102.50 | 513.643 | Surficial | Sand/Gravel | 511.181 | Loam |
| Agrium 43 | 52.03 | -107.01 | 500.229 | Intertill | Sand | 510.771 | Loam |
| 460120097591803 | 46.02 | -97.98 | 401.177 | Alluvial | Sand/Gravel | 400.381 | Sandy loam |
| 461838097553402 | 46.31 | -97.92 | 401.168 | - | Sand/Gravel | 404.719 | Clay loam |
| 462400097552502 | 46.39 | -97.92 | 409.73 | - | Sand/Gravel | 407.405 | Sandy loam |
| 462633097163402 | 46.44 | -97.27 | 325.52 | Alluvial | Sand/Gravel | 323.728 | Sandy loam |
| 463422097115602 | 46.57 | -97.19 | 320.40 | Alluvial | Sand/Gravel | 314.167 | Sandy loam |
| 464540100222101 | 46.76 | -100.37 | 524.91 | - | Sand/Gravel | 522.600 | Clay loam |
| 473841096153101 | 47.64 | -96.25 | 351.77 | Surficial | Sand/Gravel | 344.180 | Loamy sand |
| 473945096202402 | 47.66 | -96.34 | 327.78 | Surficial | Sand/Gravel | 328.129 | Sandy loam |
| 474135096203001 | 47.69 | -96.34 | 325.97 | Surficial | Sand/Gravel | 327.764 | Sandy loam |
| 474436096140801 | 47.74 | -96.23 | 341.90 | Surficial | Sand/Gravel | 336.210 | Sandy loam |
| 475224098443202 | 47.87 | -98.74 | 451.33 | - | Sand/Gravel | 450.463 | Sandy loam |
| 481841097490301 | 48.31 | -97.81 | 355.61 | - | Sand/Gravel | 359.568 | Clay loam |
| 482212099475801 | 48.37 | -99.79 | 488.65 | - | Sand/Gravel | 488.022 | Sandy loam |
| CRN Well WLN03 | 45.98 | -95.20 | 410.7 | Surficial | Sand/Gravel | 411.4 | Sandy loam |


**Table 2.** Summary of mean and standard deviation (std) of WTD from 33 groundwater wells, from
observation records (OBS), default model (CTRL) and replacing with sand soil simulation (REP).
Bold texts indicate improvement in the REP than the CTRL run.

| Site Name/Number | OBS_mean | CTRL_mean | REP_mean | OBS_std | CTRL_std | REP std |
|---|---|---|---|---|---|---|
| Devon 0162 | -2.46 | -2.69 | **-2.38** | 0.43 | 0.45 | 0.09 |
| Hardisty 0143 | -2.44 | -8.91 | **-6.88** | 0.41 | 0.64 | **0.36** |
| Kirkpatrick Lake 0229 | -4.22 | -4.03 | -3.45 | 0.43 | 0.98 | **0.22** |
| Metiskow 0267 | -2.54 | -5.39 | **-4.43** | 0.34 | 0.78 | **0.55** |
| Narrow Lake 252 | -2.31 | -4.81 | **-3.75** | 0.28 | 0.60 | 0.51 |
| Wagner 0172 | -2.14 | -8.06 | **-2.70** | 0.48 | 0.37 | 0.21 |
| Baildon 060 | -2.80 | -3.29 | **-3.20** | 0.47 | 0.58 | 0.30 |
| Beauval | -3.78 | -4.85 | **-4.20** | 0.44 | 0.56 | 0.32 |
| Blucher | -2.20 | -4.24 | **-2.16** | 0.3 | 0.92 | **0.26** |
| Crater Lake | -4.33 | -3.97 | -3.64 | 1.1 | 0.4 | 0.28 |
| Duck Lake | -3.65 | -3.69 | -3.17 | 0.54 | 0.41 | **0.62** |
| Forget | -2.28 | -2.37 | **-2.23** | 0.33 | 0.17 | 0.19 |
| Garden Head | -3.67 | -4.85 | **-3.77** | 0.88 | 0.70 | 0.30 |
| Nokomis | -1.04 | -2.70 | **-2.17** | 0.23 | 0.55 | **0.17** |
| Shaunavon | -1.62 | -4.41 | **-2.58** | 0.42 | 0.69 | 0.20 |
| Simpson 13 | -4.82 | -4.83 | -3.02 | 0.31 | 0.91 | **0.17** |
| Simpson 14 | -2.03 | -2.61 | **-1.82** | 0.34 | 0.18 | **0.27** |
| Yorkton 517 | -2.87 | -3.97 | **-1.98** | 0.8 | 0.46 | 0.32 |
| Agrium 43 | -2.66 | -3.75 | **-3.38** | 0.32 | 1.05 | **0.36** |
| 460120097591803 | -1.44 | -2.33 | **-1.63** | 0.56 | 0.24 | **0.50** |
| 461838097553402 | -1.17 | -2.32 | **-1.68** | 0.27 | 0.24 | 0.43 |
| 462400097552502 | -4.9 | -5.61 | **-5.37** | 0.29 | 0.09 | **0.17** |
| 462633097163402 | -1.18 | -1.49 | **-1.02** | 0.46 | 0.29 | **0.54** |
| 463422097115602 | -1.36 | -2.28 | **-1.66** | 0.34 | 0.23 | 0.49 |
| 464540100222101 | -2.02 | -3.64 | **-2.78** | 0.52 | 0.43 | 0.32 |
| 473841096153101 | -0.77 | -1.48 | **-1.37** | 0.24 | 0.18 | 0.51 |
| 473945096202402 | -1.59 | -1.58 | -1.56 | 0.32 | 0.24 | 0.51 |
| 474135096203001 | -0.72 | -1.48 | **-1.30** | 0.33 | 0.25 | 0.54 |
| 474436096140801 | -2.44 | -2.29 | -1.96 | 0.39 | 0.21 | **0.40** |
| 475224098443202 | -4.52 | -4.28 | -5.31 | 0.75 | 0.52 | 0.34 |
| 481841097490301 | -4.39 | -4.24 | -4.58 | 0.79 | 0.28 | 0.17 |
| 482212099475801 | -2.13 | -2.32 | **-2.26** | 0.24 | 0.20 | 0.17 |
| CRN WLN 03 | -2.04 | -2.18 | -1.88 | 0.24 | 0.18 | 0.43 |


(a)                                    (b)

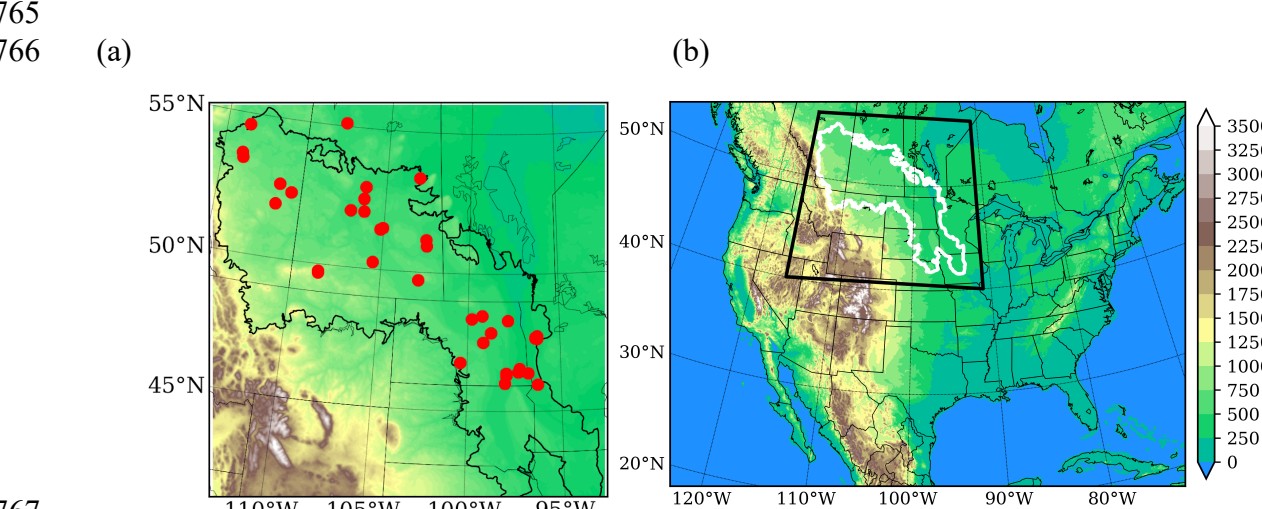

**Fig. 1** (a) Topography of the Prairie Pothole Region (PPR; black outline) and groundwater wells (red dots); (b) Topography
of the WRF CONUS domain, the black box indicates the PPR domain.


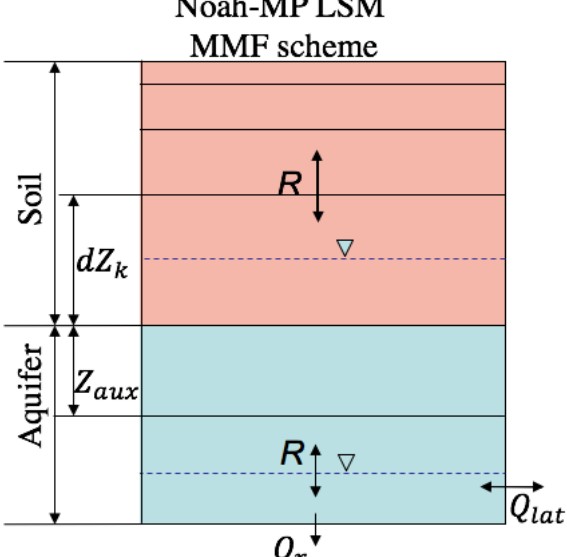


**Fig. 2** Structure of the Noah-MP LSM coupled with MMF groundwater scheme, the top 2-m soil of 4 layers whose thicknesses
are $0.1, 0.3, 0.6$ and $1.0$ m. An unconfined aquifer is added below the 2-m boundary, including an auxiliary layer and the saturated
aquifer. Positive flux of $R$ denotes downward flow. Two water tables are shown, one within the 2-m soil and one below,
indicating that the model is capable to deal with both shallow and deep water table.

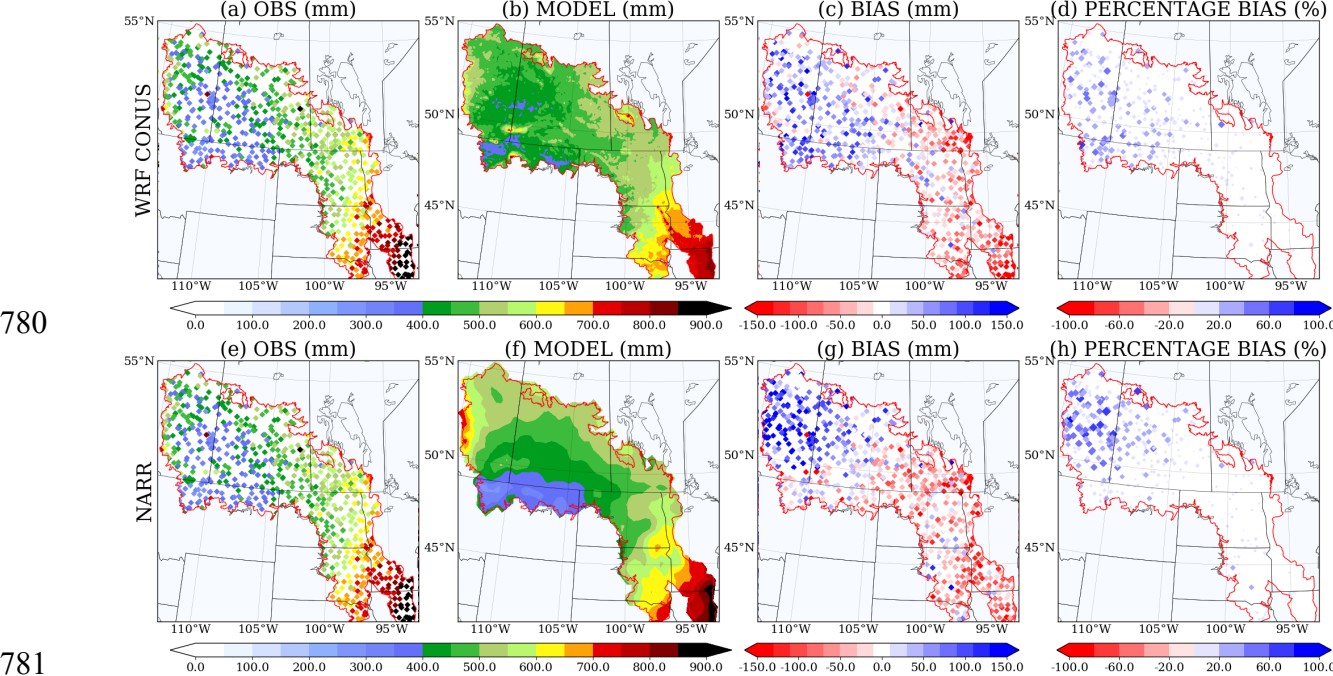


**Fig. 3** Evaluation of the annual precipitation from two model products (b, f), WRF CONUS and NARR against
rain gauge observation (a, e), their bias (c, g) and percentage bias (d, h).

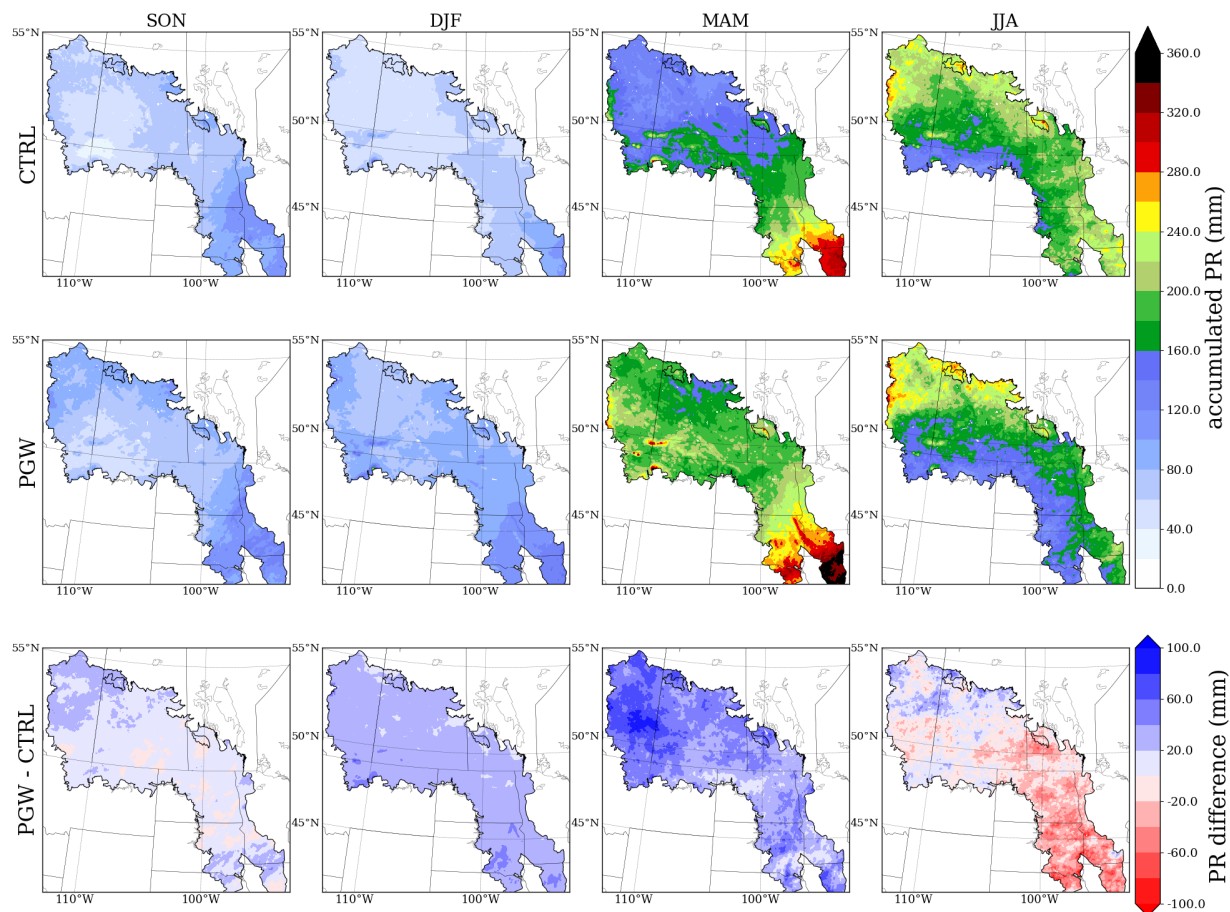

**Fig. 4** Seasonal Accumulated precipitation from current climate (CTRL, top), future climate (PGW, middle) and
projected change (PGW-CTRL, bottom) in forcing data.

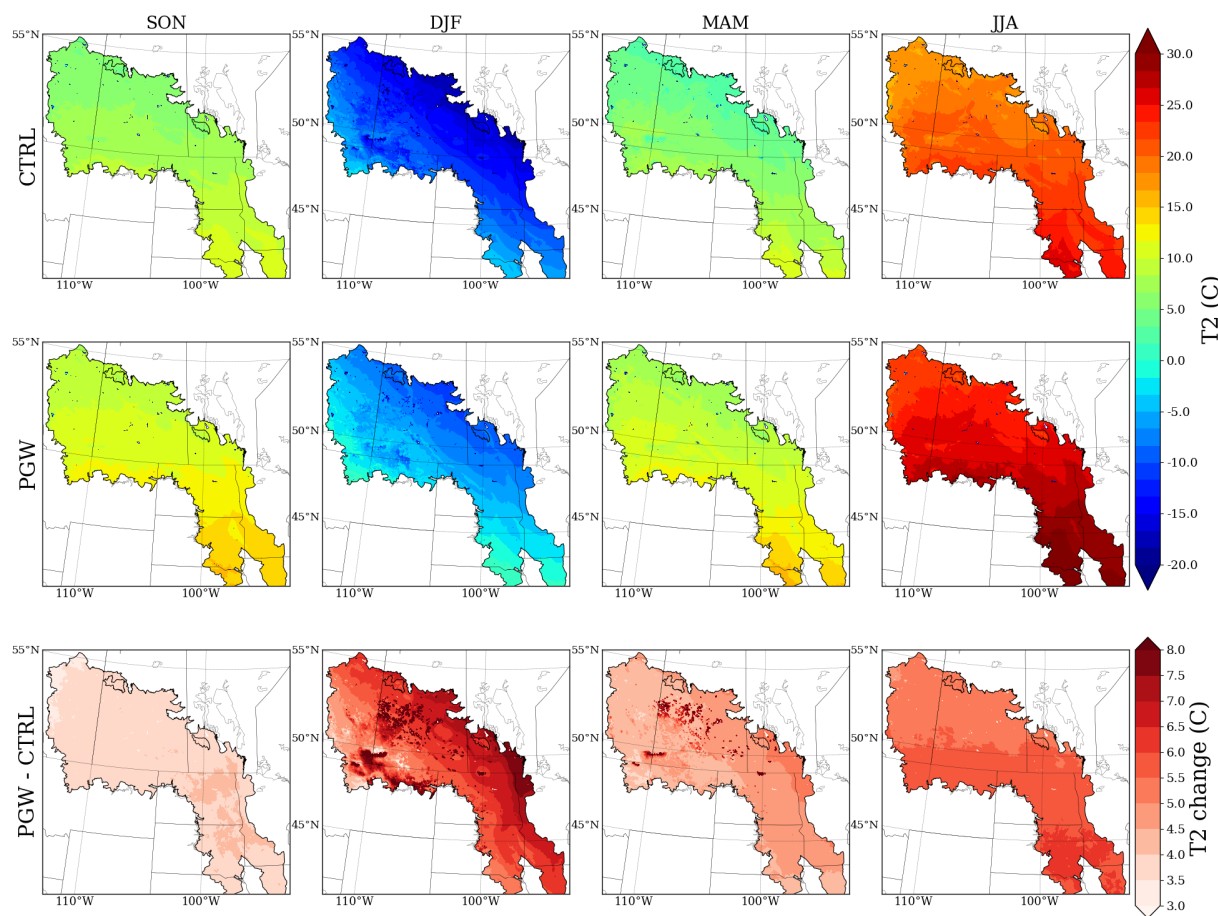

**Fig. 5** Seasonal temperatures from current climate (CTRL, top), future climate (PGW, middle) and projected
change (PGW-CTRL, bottom) in forcing data.


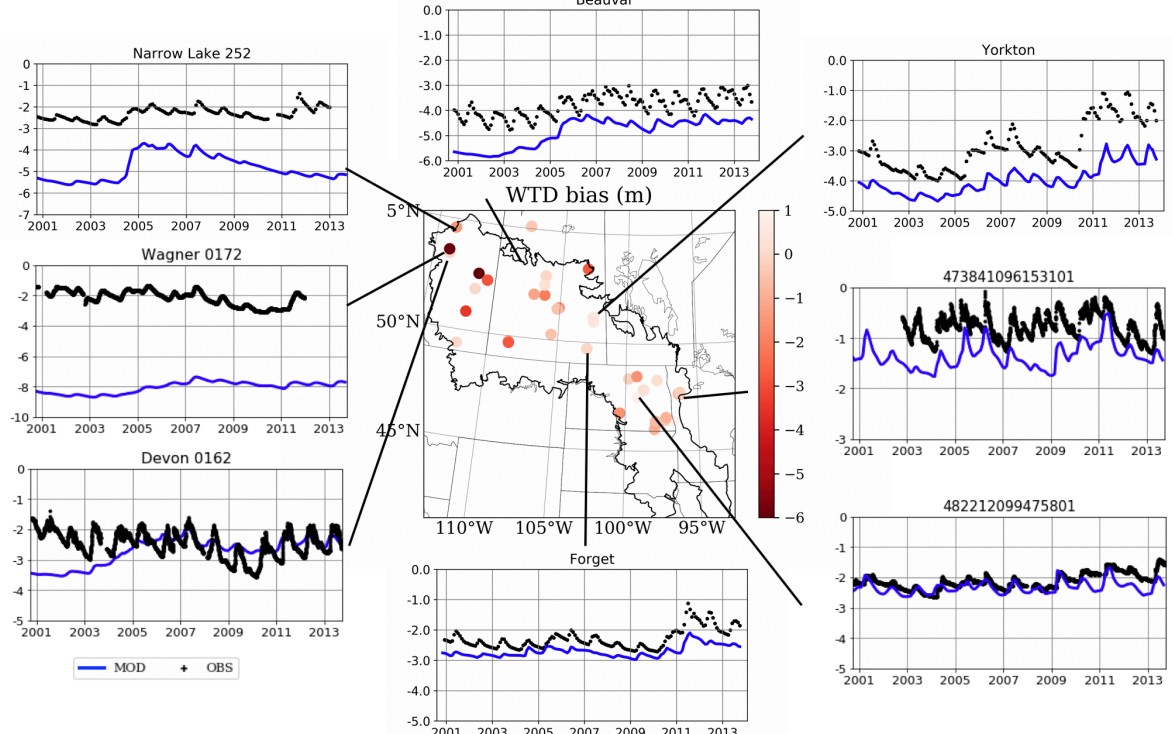

**Fig. 6.** WTD (m) bias from CTRL simulation and timeseries from 8 groundwater wells in PPR (black for observation and
blue for CTRL model simulation). See Table 2 CTRL column for the model statistics and supplemental materials for complete
timeseries from 33 wells.

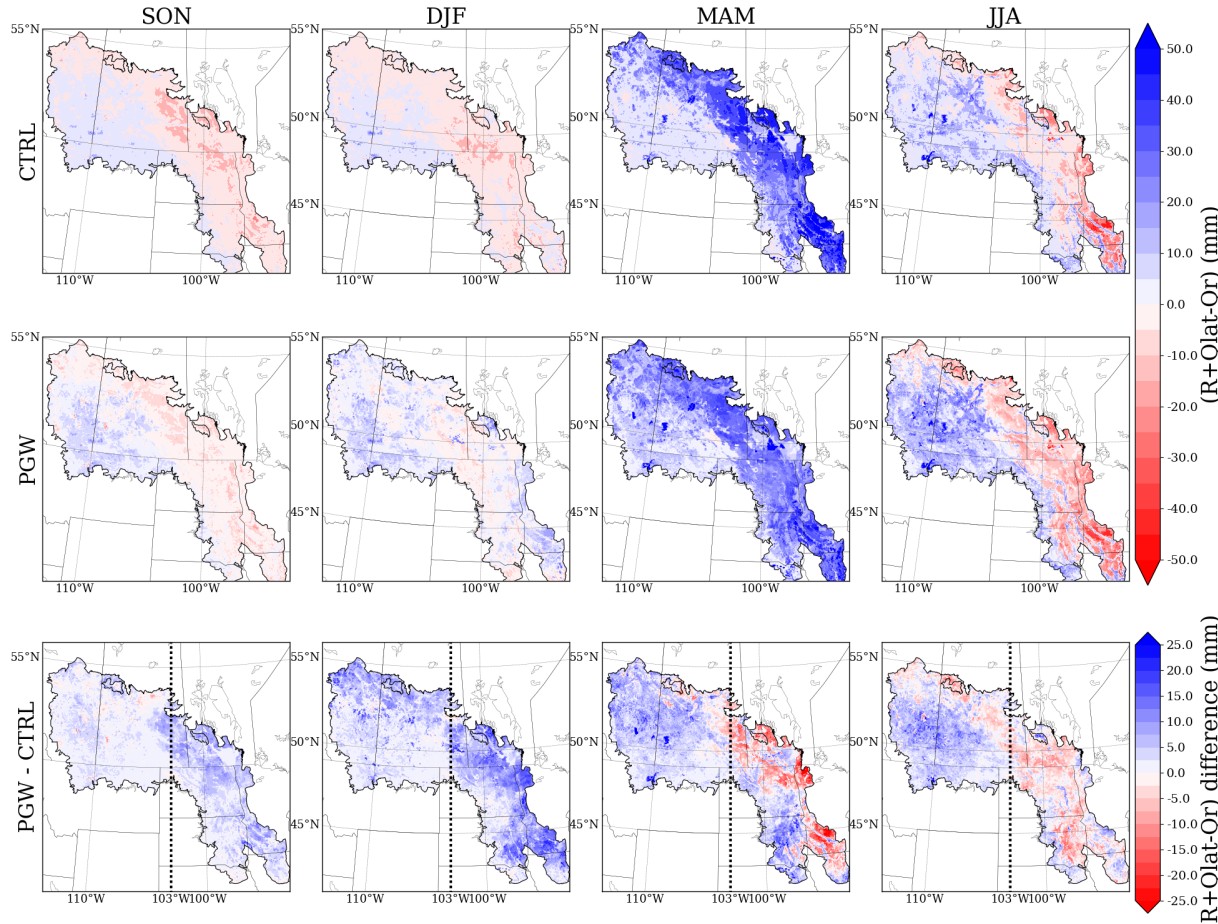

**Fig. 7** Seasonal accumulated total groundwater fluxes ($R+Q_{lat} - Q_r$) for current climate (CTRL, top), future
climate (PGW, middle) and projected change (PGW-CTRL, bottom) in forcing data. Black dashed lines in PGW-
CTRL separate the PPR into eastern and western halves.

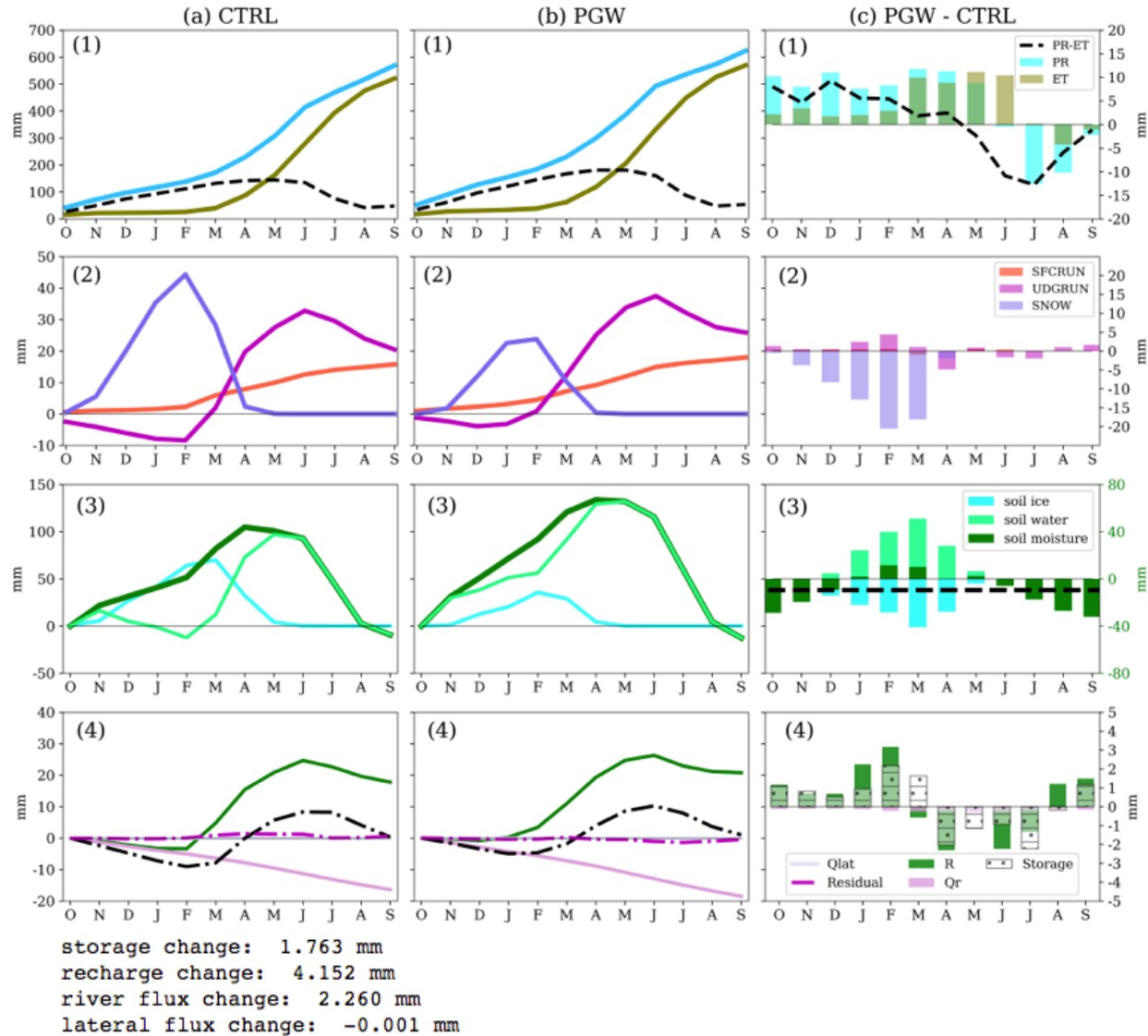


storage change:    1.763 mm
recharge change:    4.152 mm
river flux change:    2.260 mm
lateral flux change:    -0.001 mm

**Fig. 8** Water budget analysis in the **eastern PPR** in (a) CTRL, (b) PGW and (c) PGW – CTRL. Water budget terms
include: (1) *PR & ET*, (2) surface snow, surface runoff and underground runoff (*SNOW*, *SFCRUN*, and *UDGRUN*),
(3)  change of soil moisture storage (soil water, soil ice and total soil moisture, $\Delta SMC$) and (4) groundwater fluxes
and the change of groundwater storage ($R$, $Q_{lat}$, $Q_r$, $\Delta S_g$). The annual mean soil moisture change (PGW-CTRL) is
shown with black dashed line in (3). The Residual term is defined as *Res* = ($R+Q_{lat}-Q_r$)-$\Delta S_g$ in (4). Note that in (a)
and (b) the accumulated fluxes and change in storage are shown in lines, whereas in (c) the difference in (PGW-CTRL)
is shown for each individual month in bars.

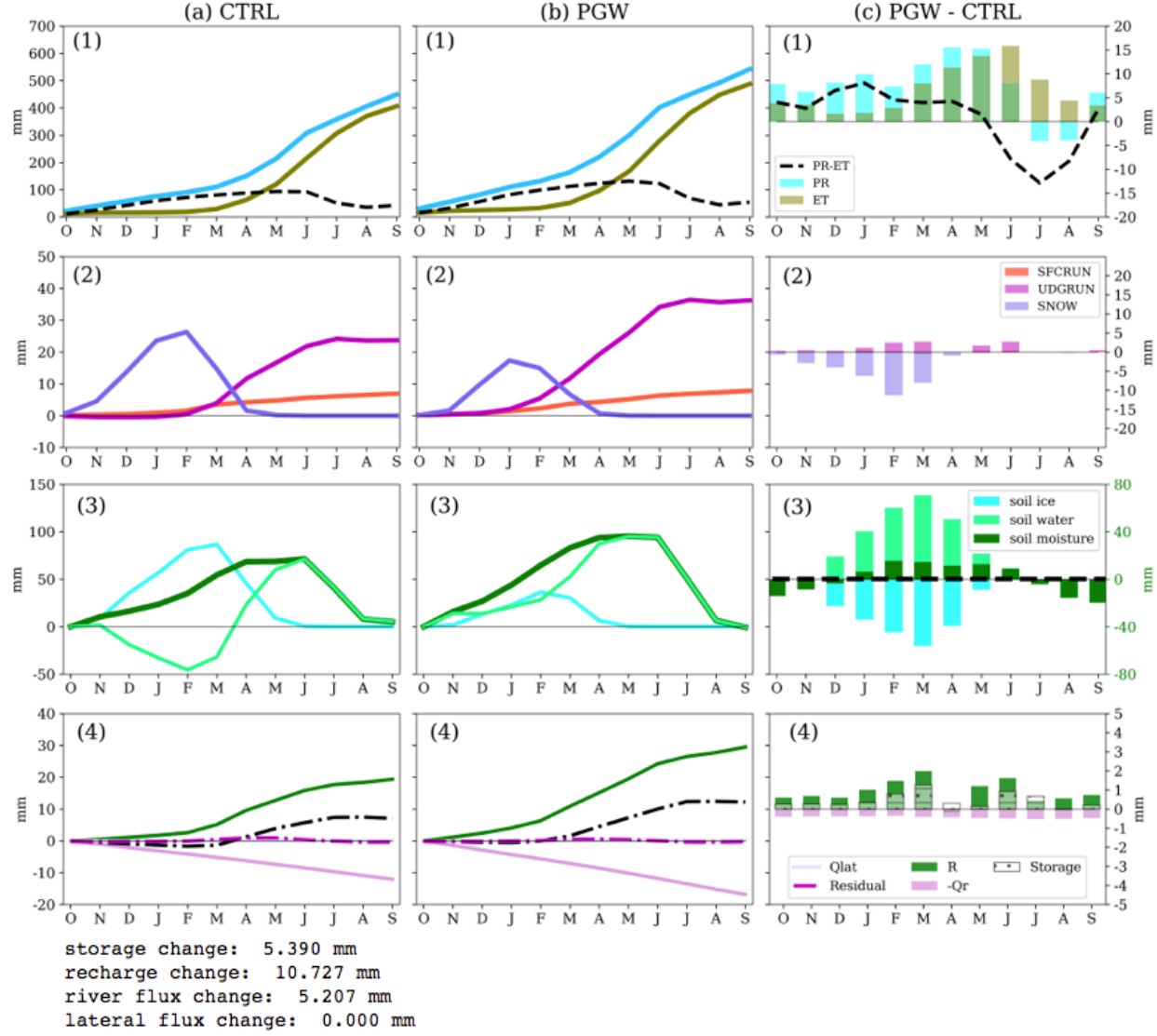

storage change:    5.390 mm
recharge change:   10.727 mm
river flux change:  5.207 mm
lateral flux change:  0.000 mm

**Fig. 9** Same as Fig. 8. Water budget analysis in the **western PPR**: in (a) CTRL, (b) PGW and (c) PGW – CTRL. Water budget terms include: (1) *PR & ET*, (2) surface snow, surface runoff and underground runoff (*SNOW*, *SFCRUN*, and *UDGRUN*), (3)  change of soil moisture storage (soil water, soil ice and total soil moisture, $\Delta SMC$) and (4) groundwater fluxes and the change of groundwater storage ($R$, $Q_{lat}$, $Q_r$, $\Delta S_g$). The annual mean soil moisture change (PGW-CTRL) is shown with black dashed line in (3). The Residual term is defined as $Res = (R+Q_{lat}-Q_r)-\Delta S_g$ in (4). Note that in (a) and (b) the accumulated fluxes and change in storage are shown in lines, whereas in (c) the difference in (PGW-CTRL) is shown for each individual month in bars.

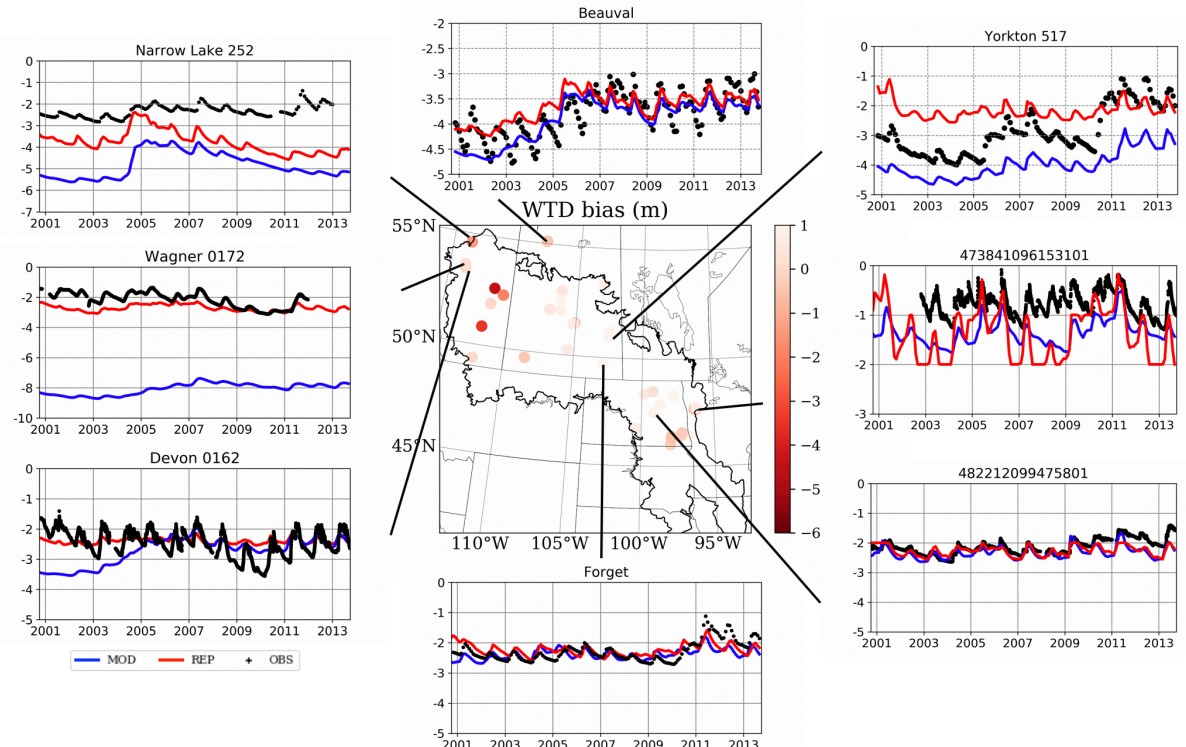

**Fig. 10** Same as Fig. 6, WTD (m) bias from CTRL simulation and timeseries from 8 groundwater wells in PPR (black for
observation and blue for CTRL model simulation, and red for the replacing soil type simulation). REP is the additional
simulation by replacing the default soil type in the model with sandy soil type.