# Peer review of "Modeling groundwater responses to climate change in the Prairie Pothole Region Zhe Zhang1, Yanping Li11, Michael Barlage2, Fei Chen2, Gonzalo Miguez-Macho3, Andrew Ireson1, Zhenhua Li Global Institute for Water Security, University of"

_Hydrology and Earth System Sciences, 2019_

## Referee Comment (RC1) · Brian Smerdon (Referee) · 6 Jun 2019

General Comments

This manuscript describes a land surface model linked to a basic groundwater model to investigate water table depth across the Prairie Pothole Region of North America. The coupled model is first used to represent recent conditions (2000 to 2013), then a future climate scenario. The manuscript addresses a relevant question regarding the hydrology of a large region, and presents a method that would be applicable in other regions. The findings are interesting and would be of interest to researchers working in smaller areas within the Prairie Pothole Region. The approach demonstrates how cold region processes can be considered in large scale models that include a basic

groundwater component.

Specific Comments

There are 3 specific comments that warrant more attention:

1) One finding of the study is that simulated water table depth is sensitive to parameterization of the soil properties, which were input from a global dataset. The authors indicate that replacing some of the default soil type parameters with more location-specific information improves the match between simulated and observed water table depth (lines 448-453). This is great to see; however, there is no indication of the difference to the future climate scenario and water budget. The net effect on the primary question (i.e. future climate) is needed for completion. How much does the REP approach change the conclusions regarding distribution of recharge under the future climate? Addressing this issue would help the authors convey how important the fine-scale properties might be.

2) The culling criteria for groundwater observation data may have been a bit ruthless. To end up with only 7% of the potential observation data seems quite aggressive. Whilst I don't disagree with the culling criteria, it would be helpful to have some additional points for spatial coverage that could be considered a "secondary" dataset (e.g. reported as supplemental material). To better understand (and accept) the culling procedure, some additional details are needed in lines 164-166. What is meant by a "sufficiently longrecord"? (provide an example or specify the timeframe). How were anthropogenic effects considered? Why was 7m selected as a cut off? Relaxing these criteria even just a little will increase the spatial coverage of your observation data.

3) Related to the culling criteria, we evaluated the Alberta groundwater observation well data in a comparison with GRACE (Huang et al. 2016, Hydrogeology Journal v24, 1663-1680). You might be able to use Table 1 from that paper to increase the number of observation records. Also, you might be able to incorporate the GRACE comparison to water level data into your discussion and conclusions.

Technical Corrections

L17: Typo "on groundwater recharge rates"

L21: Is "mismatch" really the correct term? What you're describing is a parameter that is not represented in the model adequately. The resultant water table depth is mismatched, but the parameter is misrepresented.

L23: Type "delaying the time..."

L47: A reference for the general concept of recharge and frozen soil would be useful (e.g. Hayashi)

L61: Typo "discharge to rivers"

L64-65: Suggested edit "...snowmelt recharge to reach the water table, the previously upward water movement by capillary effect to reverse and move downwards, and allow the water table to rise to..."

L66: Suggest removing "and desiccates the soil", as this starts to invoke ideas of seasonally varying parameters.

L76: Provide a reference for the 5-40mm/yr example.

L80: Typo "...this is challenging to represent in current..."

L85: Typo "suggested"

L97: Typo "groundwater models"

L128: What is meant by "groundwater evolution"? Do you simply mean water table dynamics?

L143: Typo "...from the WRF..."

L148: You might want to mention that most of the observation well data will be biased toward more permeable deposits (e.g. sand and gravel). Typically, provincial and state

agencies don't monitor low permeability formation.

L153: Alberta Environment and Parks

L164: Provide an example timeframe

L165: How was anthropogenic effect determined?

L259: Add "in the PPR" to the end of this sentence

L285-288: The model initialization process is unclear. Spin up times of 500 years and 4 years are mentioned here. Is the 4 yr period simply to account for grid size difference, and essentially following the 500 yr spin up in the previous model?

L317-318: Relation to Amazon with is not relevant and totally looks like self-citation here. The concept of infiltration response is pretty basic.

L321: By "out-of-the-box" do you simply mean "uncalibrated"?

L323: Instead of "further study" do you mean "preliminary study", because later in the manuscript modified parameters are used (i.e. REP) Section 3.2 and 3.3: A little bit of set-up is needed here. Are the results presented averages over a certain period? What timeframe for the water budget components correspond to? (3 months)

L425: Typo "...in the locations of the observations well."

L448-453: This section kinda teases the reader. How do the future scenario results look with the REP parameters?

L520: Typo "As a result..."

---

## Referee Comment (RC2) · Katie Markovich (Referee) · 11 Jun 2019

Zhang et al. use a land surface model coupled to a two-way groundwater dynamics model to explore the response of groundwater to climate change in the Prairie Pothole Region (PPR) of North America. The main research objectives of this study were to simulate two-way exchange in the subsurface, characterize groundwater response to climate change, and identify the major processes controlling this response in the region. The novel methodological components of this study include the application of a two-way groundwater exchange module coupled to a land surface model and the use of regional scale WRF CONUS model outputs with a scheme that treats convective precipitation. The authors point out that there is a need to explore hydrologic response to climate change at the regional scale, of which there is currently a gap, and so this

study is a timely and important addition to that area of research.

My main comments are:

1. The authors need to introduce their research objective earlier in the introduction, and better organize the following paragraphs around the background and motivation for the study. In particular, I would like to see more motivation for the methodological set up of the study. For example, if prairie pothole hydrogeology is so complex at the local scale (lines 70-83), why is a coarse regional model appropriate? Second, since the freeze thaw process is a key requirement for PPR hydrogeology, I would like to see more background (including references) of how this process has been treated in various LSMs and also more information in the methods section of how NOAH-MP represents it.

2. More information is needed on the criteria for selecting wells, such as required length of record and how anthropogenic effects were minimized. Further, comparing a coarse land surface model covering a large total area (the model area is not reported) to only 11 wells is concerning. From Fig. 1 it appears that quite a large portion of the PPR does not have any well coverage. I think it would be worth revisiting the criteria and including a few more wells out of the 160. If that's not possible, then more discussion regarding potential uncertainty in capturing groundwater dynamics in PPR subregions without wells (e.g. the southwest portion) is warranted.

3. More information is needed on the climate change scenarios such as what emissions scenario was used and what sort of temperature increase does that roughly translate to?

4. If the REP model performed better against observational data, then why didn't the authors choose to run the climate change scenario using that parameterization? Including such a simulation would give the reader a sense of how sensitive projections are to model parameterization. If including this simulation is too computationally expensive, then at least some discussion of how the climate projections might be sensitive to

model parameterization is warranted.

5. There is a fairly major typographical error in the text and figure captions. Delta S should be equal to R + Qlat – Qr (according to equation 4), but it is repeatedly written as R + Qlat + Qr (equation 10, for example) in the paper. This is hopefully only typographical, as that would result in large errors in reported changes in storage.

6. The figure captions in the text often do not match the figure captions associated with the figures. Further, the authors should write out fully descriptive figure captions, including defining acronyms, such that the figures would be able to be read on their own. There are currently several figure captions that simply say "same as Fig. xx."

7. Finally, the timing and amount of thaw is a key control on recharge projections, but the authors do not explore or discuss how well their model captures freeze-thaw dynamics at the regional scale. This is related to my earlier comment, that the authors need to explain how this process is represented in their model. Some discussion of how future studies could improve upon this methodology to capture this important and heterogeneous process would also add strength to the paper.

Specific comments:

There were quite a number of typographical errors, a few of which I will list here, but I do recommend the authors go back over the manuscript with a closer eye for spelling and grammatical errors.

Line 55: use a different acronym for precipitation- PR is too close to PPR

Lines 64-65: rewrite for grammar

Line 76: provide citation for recharge estimate

Line 77: rewrite for grammar

Line 80: rewrite for grammar

Lines 93-94: studies of regional climate change impacts to hydrology in N. America: Niraula et al., 2017a,b, Christensen et al., 2004

Line 148: Observational data

Line 181: define offline mode

Lines 341 and 350: under current/future climate conditions

Line 372: what is Qdrain?

Line 474: include citation

Lines 520-522: rewrite for grammar

Table 1: either use descriptive column names or define any abbreviation used. Add units where needed.

[Figure]

---

## Author Comment (AC2) · 30 Aug 2019

**Response to Reviewer #2**
(hess-2019-155)

Zhe Zhang, Yanping Li, Michael Barlage, Fei Chen, Gonzalo Miguez-Macho, Andrew Ireson, and Zhenhua Li

*Zhang et al. use a land surface model coupled to a two-way groundwater dynamics model to explore the response of groundwater to climate change in the Prairie Pothole Region (PPR) of North America. The main research objectives of this study were to simulate two-way exchange in the subsurface, characterize groundwater response to climate change, and identify the major processes controlling this response in the region. The novel methodological components of this study include the application of a two-way groundwater exchange module coupled to a land surface model and the use of regional scale WRF CONUS model outputs with a scheme that treats convective precipitation. The authors point out that there is a need to explore hydrologic response to climate change at the regional scale, of which there is currently a gap, and so this study is a timely and important addition to that area of research.*

We appreciate the editor and two reviewers. They have put into time and effort to help us improve this article and their comments and suggestions are supportive and helpful. In the following text, the general response will be in red, original reviewers' comments and questions in black and our answers to the questions in blue.

*My main comments are:*
*1. The authors need to introduce their research objective earlier in the introduction, and better organize the following paragraphs around the background and motivation for the study. In particular, I would like to see more motivation for the methodological set up of the study. For example, if prairie pothole hydrogeology is so complex at the local scale (lines 70-83), why is a coarse regional model appropriate? Second, since the freeze thaw process is a key requirement for PPR hydrogeology, I would like to see more background (including references) of how this process has been treated in various LSMs and also more information in the methods section of how NOAH-MP represents it.*
Thank you for the comment. This comment is very good and contains several questions. We would break this down and answer it separately.

The order of the paragraphs has been reorganized in the manuscript. The objectives of this study have been moved ahead in Introduction. The prairie pothole wetlands and groundwater-wetland exchanges is now moved to Discussion. More details for the methodological set-up, such as frozen soil treatments in LSMs, have been added in Data and Methods.

We move L70-83 to the Discussion section. This paragraph in the original manuscript introduced important ecosystem services provided by prairie pothole wetlands and the interactions between groundwater flow and these wetlands. Fine-scale processes (usually from 10 to 100 m), such as snow drift, runoff fill-spill and groundwater recharge/discharge to wetlands, have complicated the water balance in these wetlands and are challenging to LSMs (usually from 1 to 100 km) (Hayashi et al). In this work, we configured our model domain at 4-km resolution, and we acknowledged that it is insufficient to resolve these fine-scale processes. This is a current limit and future

challenge in this study area. We are currently developing sub-grid parameterization scheme to represent the presence of these fine-scale prairie pothole wetlands to the hydrological cycle, including ET and sub-surface flows. Please see the Discussion section for more details.

We add a paragraph in Introduction to provide background of frozen soil parameterizations in most common LSMs, such as CLM, NoahV3, and Noah-MP. Additionally, a description of Noah-MP frozen soil parameterizations is included in the Methods section. There are two options in Noah-MP for frozen soil permeability; option 1 is the default option in Noah-MP LSM and is adapted from Niu and Yang (2006); option 2 is inherited the Koren et al. (1999) scheme from NoahV3. We used the option 1 in our simulation. The option 1 assumes that a model grid cell consists of permeable and impermeable areas and thus uses the total soil moisture to compute hydraulic properties of the soil. The option 2 uses only the liquid water volume to calculate hydraulic properties. Additionally, option 1 assumes that soil ice has a linear (smaller) effect on infiltration, generally simulates more permeable frozen soil than option 2, which assumes soil ice has a non-linear (greater) effect on soil permeability (Niu et al., 2011). For this reason, the option 1 allows the soil water to move and redistribute more easily within the frozen soil.

*2. More information is needed on the criteria for selecting wells, such as required length of record and how anthropogenic effects were minimized. Further, comparing a coarse land surface model covering a large total area (the model area is not reported) to only 11 wells is concerning. From Fig. 1 it appears that quite a large portion of the PPR does not have any well coverage. I think it would be worth revisiting the criteria and including a few more wells out of the 160. If that's not possible, then more discussion regarding potential uncertainty in capturing groundwater dynamics in PPR sub-regions without wells (e.g. the southwest portion) is warranted.*

Thank you for this comment and your concern about selecting observation wells. We have revisited the groundwater well observations and our selecting criteria in this revision. We use the daily water table depth records from total 160 groundwater wells in the domain, including 72 from the USGS, 43 from Alberta Environment and Parks, and 45 from Saskatchewan Water Security Agency. The locations of these 446 wells are shown in Fig. S1, together with the mean WTD and the availability of the observational records within the simulation period, respectively. (The model domain contains 401 x 396 grid points and is added in the Methods section).

We revisited the criteria to select these groundwater wells: (1) the location of the well is close to the PPR region; (2) a sufficiently long record during the simulation period. We define the observation availability as the available observation period within the 13-year simulation period and select wells with observation availability greater than 80%; (3) unconfined aquifer with shallow groundwater (mean WTD > -5 m); and (4) has little anthropogenic influence.

After these culling processes, 33 wells are selected, including 6 from Alberta, 13 from Saskatchewan, and 14 from the U.S., and the locations of these 33 wells are shown in Fig. S1c and their information in Table S1. Table S2 also provides the statistics, including mean and standard deviation of WTD, for these 33 sites, from observation and our groundwater model. The complete timeseries of these 33 sites are shown at the end of this response.

[Figure]

Fig. S1. The locations of the 160 groundwater wells in the PPR region and their (a) mean WTD values; (b) observation record availability; (c) the locations of 33 groundwater wells that have shallow groundwater level and long observation record (> 80%). A complete list of their information is presented in Table S1.

Table S1. Information about the selected 33 wells in the Prairie Pothole Region.

| Site Name/ Site No. | Lat | Lon | Elevation | Aquifer type | Aquifer Lithology | Model Elevation | Model Soil type |
|---|---|---|---|---|---|---|---|
| Devon 0162 | 53.41 | -113.76 | 700.0 | Unconfined | Sand | 697.366 | Sandy loam |
| Hardisty 0143 | 52.67 | -111.31 | 622.0 | Unconfined | Gravel | 633.079 | Loam |
| Kirkpatrick Lake 0229 | 51.95 | -111.44 | 744.5 | Semi-confined | Sandstone | 778.311 | Sandy loam |
| Metiskow 0267 | 52.42 | -110.60 | 677.5 | Unconfined | Sand | 679.516 | Loamy sand |
| Wagner 0172 | 53.56 | -113.82 | 670.0 | Surficial | Sand | 670.845 | Silt loam |
| Narrow Lake 252 | 54.60 | -113.63 | 640.0 | Unconfined | Sand | 701.0 | Clay loam |
| Baildon 060 | 50.25 | -105.50 | 590.184 | Surficial | - | 580.890 | Sandy loam |
| Beauval | 55.11 | -107.74 | 434.3 | Intertill | Sand | 446.5 | Sandy loam |
| Blucher | 52.03 | -106.20 | 521.061 | Intertill | Sand/Gravel | 523.217 | Loam |
| Crater Lake | 50.95 | -102.46 | 524.158 | Intertill | Sand/Gravel/Clay | 522.767 | Loam |
| Duck Lake | 52.92 | -106.23 | 502.920 | Surficial | Sand | 501.729 | Loamy sand |
| Forget | 49.70 | -102.85 | 606.552 | Surficial | Sand | 605.915 | Sandy loam |
| Garden Head | 49.74 | -108.52 | 899.160 | Bedrock | Sand/Till | 894.357 | Clay loam |
| Nokomis | 51.51 | -105.06 | 516.267 | Bedrock | Sand | 511.767 | Clay loam |
| Shaunavon | 49.69 | -108.50 | 896.040 | Bedrock | Sand/Till | 900.433 | Clay loam |
| Simpson 13 | 51.45 | -105.18 | 496.620 | Surficial | Sand | 493.313 | Sandy loam |
| Simpson 14 | 51.457 | -105.19 | 496.600 | Surficial | Sand | 493.313 | Sandy loam |
| Yorkton 517 | 51.17 | -102.50 | 513.643 | Surficial | Sand/Gravel | 511.181 | Loam |
| Agrium 43 | 52.03 | -107.01 | 500.229 | Intertill | Sand | 510.771 | Loam |
| 460120097591803 | 46.02 | -97.98 | 401.177 | Alluvial | Sand/Gravel | 400.381 | Sandy loam |
| 461838097553402 | 46.31 | -97.92 | 401.168 | - | Sand/Gravel | 404.719 | Clay loam |
| 462400097552502 | 46.39 | -97.92 | 409.73 | - | Sand/Gravel | 407.405 | Sandy loam |
| 462633097163402 | 46.44 | -97.27 | 325.52 | Alluvial | Sand/Gravel | 323.728 | Sandy loam |
| 463422097115602 | 46.57 | -97.19 | 320.40 | Alluvial | Sand/Gravel | 314.167 | Sandy loam |
| 464540100222101 | 46.76 | -100.37 | 524.91 | - | Sand/Gravel | 522.600 | Clay loam |
| 473841096153101 | 47.64 | -96.25 | 351.77 | Surficial | Sand/Gravel | 344.180 | Loamy sand |
| 473945096202402 | 47.66 | -96.34 | 327.78 | Surficial | Sand/Gravel | 328.129 | Sandy loam |
| 474135096203001 | 47.69 | -96.34 | 325.97 | Surficial | Sand/Gravel | 327.764 | Sandy loam |
| 474436096140801 | 47.74 | -96.23 | 341.90 | Surficial | Sand/Gravel | 336.210 | Sandy loam |
| 475224098443202 | 47.87 | -98.74 | 451.33 | - | Sand/Gravel | 450.463 | Sandy loam |
| 481841097490301 | 48.31 | -97.81 | 355.61 | - | Sand/Gravel | 359.568 | Clay loam |
| 482212099475801 | 48.37 | -99.79 | 488.65 | - | Sand/Gravel | 488.022 | Sandy loam |
| CRN Well WLN03 | 45.98 | -95.20 | 410.7 | Surficial | Sand/Gravel | 411.4 | Sandy loam |

Table S2. Statistics of mean and standard deviation of WTD for the selected 33 wells in the Prairie Pothole Region. Bold number indicates the REP run has improved results than the CTRL run.

| Site Name/Number | OBS_mean | CTRL_mean | REP_mean | OBS_std | CTRL_std | REP std |
|---|---|---|---|---|---|---|
| Devon 0162 | -2.46 | -2.69 | **-2.38** | 0.43 | 0.45 | 0.09 |
| Hardisty 0143 | -2.44 | -8.91 | **-6.88** | 0.41 | 0.64 | **0.36** |
| Kirkpatrick Lake 0229 | -4.22 | -4.03 | -3.45 | 0.43 | 0.98 | **0.22** |
| Metiskow 0267 | -2.54 | -5.39 | **-4.43** | 0.34 | 0.78 | **0.55** |
| Narrow Lake 252 | -2.31 | -4.81 | **-3.75** | 0.28 | 0.60 | 0.51 |
| Wagner 0172 | -2.14 | -8.06 | **-2.70** | 0.48 | 0.37 | 0.21 |
| Baildon 060 | -2.80 | -3.29 | **-3.20** | 0.47 | 0.58 | 0.30 |
| Beauval | -3.78 | -4.85 | **-4.20** | 0.44 | 0.56 | 0.32 |
| Blucher | -2.20 | -4.24 | **-2.16** | 0.3 | 0.92 | **0.26** |
| Crater Lake | -4.33 | -3.97 | -3.64 | 1.1 | 0.4 | 0.28 |
| Duck Lake | -3.65 | -3.69 | -3.17 | 0.54 | 0.41 | **0.62** |
| Forget | -2.28 | -2.37 | **-2.23** | 0.33 | 0.17 | 0.19 |
| Garden Head | -3.67 | -4.85 | **-3.77** | 0.88 | 0.70 | 0.30 |
| Nokomis | -1.04 | -2.70 | **-2.17** | 0.23 | 0.55 | **0.17** |
| Shaunavon | -1.62 | -4.41 | **-2.58** | 0.42 | 0.69 | 0.20 |
| Simpson 13 | -4.82 | -4.83 | -3.02 | 0.31 | 0.91 | **0.17** |
| Simpson 14 | -2.03 | -2.61 | **-1.82** | 0.34 | 0.18 | **0.27** |
| Yorkton 517 | -2.87 | -3.97 | **-1.98** | 0.8 | 0.46 | 0.32 |
| Agrium 43 | -2.66 | -3.75 | **-3.38** | 0.32 | 1.05 | **0.36** |
| 460120097591803 | -1.44 | -2.33 | **-1.63** | 0.56 | 0.24 | **0.50** |
| 461838097553402 | -1.17 | -2.32 | **-1.68** | 0.27 | 0.24 | 0.43 |
| 462400097552502 | -4.9 | -5.61 | **-5.37** | 0.29 | 0.09 | **0.17** |
| 462633097163402 | -1.18 | -1.49 | **-1.02** | 0.46 | 0.29 | **0.54** |
| 463422097115602 | -1.36 | -2.28 | **-1.66** | 0.34 | 0.23 | 0.49 |
| 464540100222101 | -2.02 | -3.64 | **-2.78** | 0.52 | 0.43 | 0.32 |
| 473841096153101 | -0.77 | -1.48 | **-1.37** | 0.24 | 0.18 | 0.51 |
| 473945096202402 | -1.59 | -1.58 | -1.56 | 0.32 | 0.24 | 0.51 |
| 474135096203001 | -0.72 | -1.48 | **-1.30** | 0.33 | 0.25 | 0.54 |
| 474436096140801 | -2.44 | -2.29 | -1.96 | 0.39 | 0.21 | **0.40** |
| 475224098443202 | -4.52 | -4.28 | -5.31 | 0.75 | 0.52 | 0.34 |
| 481841097490301 | -4.39 | -4.24 | -4.58 | 0.79 | 0.28 | 0.17 |
| 482212099475801 | -2.13 | -2.32 | **-2.26** | 0.24 | 0.20 | 0.17 |
| CRN WLN 03 | -2.04 | -2.18 | -1.88 | 0.24 | 0.18 | 0.43 |

*3. More information is needed on the climate change scenarios such as what emissions scenario was used and what sort of temperature increase does that roughly translate to?*

Thank you for this comment, we have added the information about emission scenario for the PGW forcing. The climate change forcing used in this LSM study is from a regional convective-permitting modeling project in North America, called WRF CONUS. Liu et al. (2017) discussed the WRF CONUS project in details. The CTRL run is forced with 6-hour ERA-Interim data, representing the current climate. The PGW run is forced with the ERA-Interim data plus a climate perturbation derived from CMIP5 ensemble under the RCP8.5 emission scenario, representing the future climate change till the end of 21$^{st}$ century. Fig. 4 and 5 in the manuscript show the temperature and precipitation change in PGW-CTRL. The most significant warming occurs in the winter over northern region and mountainous region in Alberta, warming up to 8 °C. An overall increase in precipitation is shown in Fig. 5, except in summer in the southeast, about 50 to 100 mm reduction.

*4. If the REP model performed better against observational data, then why didn't the authors choose to run the climate change scenario using that parameterization? Including such a simulation would give the reader a sense of how sensitive projections are to model parameterization. If including this simulation is too computationally expensive, then at least some discussion of how the climate projections might be sensitive to.*

We appreciate that both reviewers have asked a question about the responses of REP soil under future PGW climate forcing (PGW forcing). This is also an important point we need better elaborate in the manuscript and in this reply.

We revisit the observational groundwater wells and select 33 out of 160 wells (see Answer 2) and replace the default soil with sand in these 33 locations. For the rest of the domain, we keep the default soil type from the 1-km global soil map. The complete list of 33 groundwater observation wells and the modeled WTD with default (DEF, blue lines) soil and REP soil (red lines) are in Fig. S3 at the end of this document. We also conducted a simulation with REP soil under PGW climate. Ten sites are presented here as they show diverse results in these sites (see Fig. S2).

In general, under PGW climate, WTD rises due to increased precipitation and recharge. For some sites, the rise of WTD is more obvious in DEF soil rather than REP soil (e.g. Kirkpatrick Lake, Hardisty, Metiskow and 48184097490301). This is because the WTD under CTRL_REP is already higher than the WTD in CTRL_DEF and the $Q_r$ term (groundwater discharge to rivers) is parameterized as the gradient between WTD and riverbed (Eq. (8)). As a loss term in the groundwater flux, $Q_r$ is stronger in REP soil than in DEF soil and the climate change impacts on WTD rise is less prominent in REP soil than in DEF soil. On the other hand, there are some sites where PGW has little impacts on WTD, such as Simpson, Duck Lake and 482212099475801.

On point scale, given these diverse results over a limited number of sites, it is difficult to draw a universal conclusion but keep in mind the uncertainties and sensitivity of modeled WTD to soil parameters. On regional scale, the modifications of soil type at these 33 sites have little contribution to the large domain (401 x 396 grid points). Thus, our results of regional averaged water budget analysis in eastern and western PPR (Fig. 8 & 9) still hold. An ideal method to address this is to obtain sufficient information on soil properties accounting for horizontal and vertical

heterogeneity. This is an on-going project that we are working on with the support from the Global Water Futures project and future improvements can be expected.

[Figure]

Fig. S2, the WTD dynamics of the observation and 4 model simulations: the two blue lines for default soil type (DEF), and two red lines for REP soil type (changed from default to sand); and solid lines for current climate (CTRL) and dashed line for future climate (PGW).

*5. There is a fairly major typographical error in the text and figure captions. Delta S should be equal to R + Qlat – Qr (according to equation 4), but it is repeatedly written as R + Qlat + Qr (equation 10, for example) in the paper. This is hopefully only typographical, as that would result in large errors in reported changes in storage.*

Thank you very much for pointing out this typo mistake. The $Qr$ term characterizes the groundwater discharge to maintain river flow and in the model this term is always positive, meaning water flow from aquifer to riverbed. This is a loss term to the groundwater aquifer and that's why there is a negative sign before the term. I have corrected this in the manuscript, please see.

*6. The figure captions in the text often do not match the figure captions associated with the figures. Further, the authors should write out fully descriptive figure captions, including defining acronyms, such that the figures would be able to be read on their own. There are currently several figure captions that simply says, "same as Fig. xx."*
Thank you for the comment. Figure captions have been changed.

*7. Finally, the timing and amount of thaw is a key control on recharge projections, but the authors do not explore or discuss how well their model captures freeze-thaw dynamics at the regional scale. This is related to my earlier comment, that the authors need to explain how this process is represented in their model. Some discussion of how future studies could improve upon this methodology to capture this important and heterogeneous process would also add strength to the paper.*

Thank you very much and we appreciate this comment. In this revision, we introduced the frozen soil parameterizations in Noah-MP and other LSMs in the Introduction and Methods section. Although it is still a challenge to explore freeze-thaw dynamics on regional scale, we include some discussion on this matter and hope it can encourage future studies.

To our knowledge, there is no direct observation of soil ice for large region coverage. Most of the existing soil ice measurement are on local scale, for example, measurement from the FLUXNET sites, and have been used in for model evaluation (Niu and Yang 2006; Niu et al., 2011). Yang et al. (2011) provided a regional analysis on runoff, using the University of New Hampshire-Global Runoff Data Center dataset, and inferred the improved runoff simulation is the more permeable frozen soil in Noah-MP. These contents are also added to the Discussion as well.

*Specific comments:*
*There were quite a number of typographical errors, a few of which I will list here, but I do recommend the authors go back over the manuscript with a closer eye for spelling and grammatical errors.*

*Line 55: use a different acronym for precipitation- PR is too close to PPR*
*Lines 64-65: rewrite for grammar*
Thanks for the correction, done.

*Line 76: provide citation for recharge estimate*
Thanks for the correction, reference (Hayashi et al., 2016) added.

*Line 77: rewrite for grammar*
*Line 80: rewrite for grammar*
Done. This paragraph is now moved to Discussion.

*Lines 93-94: studies of regional climate change impacts to hydrology in N. America: Niraula et al., 2017a,b, Christensen et al., 2004*
Thanks for these citations, added in the reference list.

*Line 148: Observational data*
*Line 181: define offline mode*
*Lines 341 and 350: under current/future climate conditions*
Thank you for the correction.

*Line 372: what is Qdrain?*
Should be "negative recharge". Corrected in the manuscript.

*Line 474: include citation*
Thank you for the reminder. Reference (Pokhrel et al., 2014) added in the list.

*Lines 520-522: rewrite for grammar*
Done.

*Table 1: either use descriptive column names or define any abbreviation used. Add units where needed.*
Thank you, definitions of abbreviation are added.

Reference

Hayashi, M., van der Kamp, G. and Rosenberry, D. O.: Hydrology of Prairie Wetlands: Understanding the Integrated Surface-Water and Groundwater Processes, Wetlands, 36, 237–254, doi:10.1007/s13157-016-0797-9, 2016.

Koren, V., Schaake, J., Mitchell, K., Duan, Q.-Y., Chen, F. and Baker, J. M.: A parameterization of snowpack and frozen ground intended for NCEP weather and climate models, J. Geophys. Res. Atmos., 104(D16), 19569–19585, doi:10.1029/1999JD900232, 1999.

Mohammed, A. A., Kurylyk, B. L., Cey, E. E. and Hayashi, M.: Snowmelt Infiltration and Macropore Flow in Frozen Soils: Overview, Knowledge Gaps, and a Conceptual Framework, Vadose Zo. J., 17(1), doi:10.2136/vzj2018.04.0084, 2018.

Niu, G.-Y. and Yang, Z.-L.: Effects of Frozen Soil on Snowmelt Runoff and Soil Water Storage at a Continental Scale, J. Hydrometeorol., 7(5), 937–952, doi:10.1175/JHM538.1, 2006.

Pokhrel, Y. N., Fan, Y. and Miguez-Macho, G.: Potential hydrologic changes in the Amazon by the end of the 21st century and the groundwater buffer, Environ. Res. Lett., 9(8), doi:10.1088/1748-9326/9/8/084004, 2014.

**Supplemental Materials - WTD dynamics from 33 groundwater wells in the PPR**

Alberta Environment and Parks

[Figure]

**Saskatchewan Water Securtiy Agency**

[Figure]

[Figure]

[Figure]

USGS

[Figure]

[Figure]

Fig. S3. WTD dynamics from observational wells and CTRL model with default soil (DEF, blue lines) and replacing default soil with sandy soil (REP, red lines) for the 33 sites in the PPR.

---

## Author Response (AR1)

**Response to Reviewer #1**
**(hess-2019-155)**

Zhe Zhang, Yanping Li, Michael Barlage, Fei Chen, Gonzalo Miguez-Macho, Andrew Ireson, and Zhenhua Li

We appreciate the editor and two reviewers. They have put into time and effort to help us improve this article and their comments and suggestions are supportive and helpful. In the following text, the general response will be in red, original reviewers' comments and questions in black and our response to questions in blue.

In the REP simulation, we replaced the model default soil type, from a global 1-km resolution soil map, with the soil survey information, provided by the 11 groundwater well observations. This reviewer asked about how the REP approach change the conclusion on groundwater budget under future climate condition. As well in the comment 2, the reviewer asked about our culling criteria on selecting observation wells. The reviewer also provided us sources of groundwater wells and additional evaluation tools from GRACE satellite.

The first thing we address is to review the groundwater observation wells we selected in this study, as they provided critical information to evaluate our model output as well as soil properties to constrain our sensitivity study.

*General Comments*
*This manuscript describes a land surface model linked to a basic groundwater model to investigate water table depth across the Prairie Pothole Region of North America. The coupled model is first used to represent recent conditions (2000 to 2013), then a future climate scenario. The manuscript addresses a relevant question regarding the hydrology of a large region and presents a method that would be applicable in other regions. The findings are interesting and would be of interest to researchers working in smaller areas within the Prairie Pothole Region. The approach demonstrates how cold region processes can be considered in large scale models that include a basic groundwater component*

*Specific Comments*
*There are 3 specific comments that warrant more attention:*
*1) One finding of the study is that simulated water table depth is sensitive to parameterization of the soil properties, which were input from a global dataset. The authors indicate that replacing some of the default soil type parameters with more location-specific information improves the match between simulated and observed water table depth (lines 448-453). This is great to see; however, there is no indication of the difference to the future climate scenario and water budget. The net effect on the primary question (i.e. future climate) is needed for completion. How much does the REP approach change the conclusions regarding distribution of recharge under the future climate? Addressing this issue would help the authors convey how important the fine-scale properties might be.*

We appreciate that both reviewers have asked a question about the responses of REP soil under future PGW climate forcing (PGW forcing). This is also an important point we need better elaborate in the manuscript and in this reply.

We revisit the observational groundwater wells and select 33 out of 160 wells (see Answer 2) and replace the default soil with sand in these 33 locations. For the rest of the domain, we keep the default soil type from the 1-km global soil map. The complete list of 33 groundwater observation wells and the modeled WTD with default (DEF, blue lines) soil and REP soil (red lines) are in Fig. S4 at the end of this document. We also conducted a simulation with REP soil under PGW climate. Ten sites are presented here as they show diverse results in these sites (see Fig. S1).

[Figure]

Fig. S1, the WTD dynamics of the observation and 4 model simulations: the two blue lines for default soil type (DEF), and two red lines for REP soil type (changed from default to sand); and solid lines for current climate (CTRL) and dashed line for future climate (PGW).

In general, under PGW climate, WTD rises due to increased precipitation and recharge. For some sites, the rise of WTD is more obvious in DEF soil rather than REP soil (e.g. Kirkpatrick Lake, Hardisty, Metiskow and 48184097490301). This is because the WTD under CTRL_REP is already higher than the WTD in CTRL_DEF and the $Q_r$ term (groundwater discharge to rivers) is parameterized as the gradient between WTD and riverbed (Eq. (8)). As a loss term in the groundwater flux, $Q_r$ is stronger in REP soil than in DEF soil and the climate change impacts on WTD rise is less prominent in REP soil than in DEF soil. On the other hand, there are some sites where PGW has little impacts on WTD, such as Simpson, Duck Lake and 482212099475801.

On point scale, given these diverse results over a limited number of sites, it is difficult to draw a universal conclusion but keep in mind the uncertainties and sensitivity of modeled WTD to soil parameters. On regional scale, the modifications of soil type at these 33 sites have little contribution to the large domain (401 x 396 grid points). Thus, our results of regional averaged water budget analysis in eastern and western PPR (Fig. 8 & 9) still hold. An ideal method to address this is to obtain sufficient information on soil properties accounting for horizontal and vertical heterogeneity. This is an on-going project that we are working on with the support from the Global Water Futures project. Future results and improvements can be expected.

*2) The culling criteria for groundwater observation data may have been a bit ruthless. To end up with only 7% of the potential observation data seems quite aggressive. Whilst I don't disagree with the culling criteria, it would be helpful to have some additional points for spatial coverage that could be considered a "secondary" dataset (e.g. reported as supplemental material). To better understand (and accept) the culling procedure, some additional details are needed in lines 164-166. What is meant by a "sufficiently long record"? (provide an example or specify the timeframe). How were anthropogenic effects considered? Why was 7m selected as a cut off? Relaxing these criteria even just a little will increase the spatial coverage of your observation data.*

Thank you for this comment and your concern about selecting observation wells. We have revisited the groundwater well observations and our selecting criteria in this revision. We use the daily water table depth records from total 160 groundwater wells in the domain, including 72 from the USGS, 43 from Alberta Environment and Parks, and 45 from Saskatchewan Water Security Agency. The locations of these 160 wells are shown in Fig. S2, together with the mean WTD and the availability of the observational records within the simulation period, respectively.

We revisited the criteria to select these groundwater wells: (1) the location of the well is close to the PPR region; (2) a sufficiently long record during the simulation period. We define the observation availability as the available observation period within the 13-year simulation period and select wells with observation availability greater than 80%; (3) unconfined aquifer with shallow groundwater (mean WTD > -5 m); and (4) has little anthropogenic influence (Fig. S3 shows an example of the impacts of pumping).

After these culling processes, 33 wells are selected, including 6 from Alberta, 13 from Saskatchewan, and 14 from the U.S. The locations of these 33 wells are shown in Fig. S2c and their information in Table S1. Table S2 also provides the statistics, including mean and standard deviation of WTD, for these 33 sites, from observation and our groundwater model. The complete timeseries of these 33 sites are shown at the end of this response.

[Figure]

Fig. S2. The locations of the 160 groundwater wells in the PPR region and their (a) mean WTD values; (b) observation record availability; (c) the locations of 33 groundwater wells that have shallow groundwater level and long observation record (> 80%). A complete list of their information is presented in Table S1.

Table S1. Information about the selected 33 wells in the Prairie Pothole Region.

| Site Name/ Site No. | Lat | Lon | Elevation | Aquifer type | Aquifer Lithology | Model Elevation | Model Soil type |
|---|---|---|---|---|---|---|---|
| Devon 0162 | 53.41 | -113.76 | 700.0 | Unconfined | Sand | 697.366 | Sandy loam |
| Hardisty 0143 | 52.67 | -111.31 | 622.0 | Unconfined | Gravel | 633.079 | Loam |
| Kirkpatrick Lake 0229 | 51.95 | -111.44 | 744.5 | Semi-confined | Sandstone | 778.311 | Sandy loam |
| Metiskow 0267 | 52.42 | -110.60 | 677.5 | Unconfined | Sand | 679.516 | Loamy sand |
| Wagner 0172 | 53.56 | -113.82 | 670.0 | Surficial | Sand | 670.845 | Silt loam |
| Narrow Lake 252 | 54.60 | -113.63 | 640.0 | Unconfined | Sand | 701.0 | Clay loam |
| Baildon 060 | 50.25 | -105.50 | 590.184 | Surficial | - | 580.890 | Sandy loam |
| Beauval | 55.11 | -107.74 | 434.3 | Intertill | Sand | 446.5 | Sandy loam |
| Blucher | 52.03 | -106.20 | 521.061 | Intertill | Sand/Gravel | 523.217 | Loam |
| Crater Lake | 50.95 | -102.46 | 524.158 | Intertill | Sand/Gravel/Clay | 522.767 | Loam |
| Duck Lake | 52.92 | -106.23 | 502.920 | Surficial | Sand | 501.729 | Loamy sand |
| Forget | 49.70 | -102.85 | 606.552 | Surficial | Sand | 605.915 | Sandy loam |
| Garden Head | 49.74 | -108.52 | 899.160 | Bedrock | Sand/Till | 894.357 | Clay loam |
| Nokomis | 51.51 | -105.06 | 516.267 | Bedrock | Sand | 511.767 | Clay loam |
| Shaunavon | 49.69 | -108.50 | 896.040 | Bedrock | Sand/Till | 900.433 | Clay loam |
| Simpson 13 | 51.45 | -105.18 | 496.620 | Surficial | Sand | 493.313 | Sandy loam |
| Simpson 14 | 51.457 | -105.19 | 496.600 | Surficial | Sand | 493.313 | Sandy loam |
| Yorkton 517 | 51.17 | -102.50 | 513.643 | Surficial | Sand/Gravel | 511.181 | Loam |
| Agrium 43 | 52.03 | -107.01 | 500.229 | Intertill | Sand | 510.771 | Loam |
| 460120097591803 | 46.02 | -97.98 | 401.177 | Alluvial | Sand/Gravel | 400.381 | Sandy loam |
| 461838097553402 | 46.31 | -97.92 | 401.168 | - | Sand/Gravel | 404.719 | Clay loam |
| 462400097552502 | 46.39 | -97.92 | 409.73 | - | Sand/Gravel | 407.405 | Sandy loam |
| 462633097163402 | 46.44 | -97.27 | 325.52 | Alluvial | Sand/Gravel | 323.728 | Sandy loam |
| 463422097115602 | 46.57 | -97.19 | 320.40 | Alluvial | Sand/Gravel | 314.167 | Sandy loam |
| 464540100222101 | 46.76 | -100.37 | 524.91 | - | Sand/Gravel | 522.600 | Clay loam |
| 473841096153101 | 47.64 | -96.25 | 351.77 | Surficial | Sand/Gravel | 344.180 | Loamy sand |
| 473945096202402 | 47.66 | -96.34 | 327.78 | Surficial | Sand/Gravel | 328.129 | Sandy loam |
| 474135096203001 | 47.69 | -96.34 | 325.97 | Surficial | Sand/Gravel | 327.764 | Sandy loam |
| 474436096140801 | 47.74 | -96.23 | 341.90 | Surficial | Sand/Gravel | 336.210 | Sandy loam |
| 475224098443202 | 47.87 | -98.74 | 451.33 | - | Sand/Gravel | 450.463 | Sandy loam |
| 481841097490301 | 48.31 | -97.81 | 355.61 | - | Sand/Gravel | 359.568 | Clay loam |
| 482212099475801 | 48.37 | -99.79 | 488.65 | - | Sand/Gravel | 488.022 | Sandy loam |
| CRN Well WLN03 | 45.98 | -95.20 | 410.7 | Surficial | Sand/Gravel | 411.4 | Sandy loam |

Table S2. Statistics of mean and standard deviation of WTD for the selected 33 wells in the Prairie Pothole Region. Bold texts indicate improvement in the REP than the CTRL run.

| Site Name/Number | OBS_mean | CTRL_mean | REP_mean | OBS_std | CTRL_std | REP std |
|---|---|---|---|---|---|---|
| Devon 0162 | -2.46 | -2.69 | **-2.38** | 0.43 | 0.45 | 0.09 |
| Hardisty 0143 | -2.44 | -8.91 | **-6.88** | 0.41 | 0.64 | **0.36** |
| Kirkpatrick Lake 0229 | -4.22 | -4.03 | -3.45 | 0.43 | 0.98 | **0.22** |
| Metiskow 0267 | -2.54 | -5.39 | **-4.43** | 0.34 | 0.78 | **0.55** |
| Narrow Lake 252 | -2.31 | -4.81 | **-3.75** | 0.28 | 0.60 | 0.51 |
| Wagner 0172 | -2.14 | -8.06 | **-2.70** | 0.48 | 0.37 | 0.21 |
| Baildon 060 | -2.80 | -3.29 | **-3.20** | 0.47 | 0.58 | 0.30 |
| Beauval | -3.78 | -4.85 | **-4.20** | 0.44 | 0.56 | 0.32 |
| Blucher | -2.20 | -4.24 | **-2.16** | 0.3 | 0.92 | **0.26** |
| Crater Lake | -4.33 | -3.97 | -3.64 | 1.1 | 0.4 | 0.28 |
| Duck Lake | -3.65 | -3.69 | -3.17 | 0.54 | 0.41 | **0.62** |
| Forget | -2.28 | -2.37 | **-2.23** | 0.33 | 0.17 | 0.19 |
| Garden Head | -3.67 | -4.85 | **-3.77** | 0.88 | 0.70 | 0.30 |
| Nokomis | -1.04 | -2.70 | **-2.17** | 0.23 | 0.55 | **0.17** |
| Shaunavon | -1.62 | -4.41 | **-2.58** | 0.42 | 0.69 | 0.20 |
| Simpson 13 | -4.82 | -4.83 | -3.02 | 0.31 | 0.91 | **0.17** |
| Simpson 14 | -2.03 | -2.61 | **-1.82** | 0.34 | 0.18 | **0.27** |
| Yorkton 517 | -2.87 | -3.97 | **-1.98** | 0.8 | 0.46 | 0.32 |
| Agrium 43 | -2.66 | -3.75 | **-3.38** | 0.32 | 1.05 | **0.36** |
| 460120097591803 | -1.44 | -2.33 | **-1.63** | 0.56 | 0.24 | **0.50** |
| 461838097553402 | -1.17 | -2.32 | **-1.68** | 0.27 | 0.24 | 0.43 |
| 462400097552502 | -4.9 | -5.61 | **-5.37** | 0.29 | 0.09 | **0.17** |
| 462633097163402 | -1.18 | -1.49 | **-1.02** | 0.46 | 0.29 | **0.54** |
| 463422097115602 | -1.36 | -2.28 | **-1.66** | 0.34 | 0.23 | 0.49 |
| 464540100222101 | -2.02 | -3.64 | **-2.78** | 0.52 | 0.43 | 0.32 |
| 473841096153101 | -0.77 | -1.48 | **-1.37** | 0.24 | 0.18 | 0.51 |
| 473945096202402 | -1.59 | -1.58 | -1.56 | 0.32 | 0.24 | 0.51 |
| 474135096203001 | -0.72 | -1.48 | **-1.30** | 0.33 | 0.25 | 0.54 |
| 474436096140801 | -2.44 | -2.29 | -1.96 | 0.39 | 0.21 | **0.40** |
| 475224098443202 | -4.52 | -4.28 | -5.31 | 0.75 | 0.52 | 0.34 |
| 481841097490301 | -4.39 | -4.24 | -4.58 | 0.79 | 0.28 | 0.17 |
| 482212099475801 | -2.13 | -2.32 | **-2.26** | 0.24 | 0.20 | 0.17 |
| CRN WLN 03 | -2.04 | -2.18 | -1.88 | 0.24 | 0.18 | 0.43 |

Additionally, Fig. S2 provides an example of anthropogenic influence – pumping – which is the most common case in the PPR. The hydrograph is from a groundwater observation well in Vanscoy, SK (https://www.wsask.ca/Water-Info/Ground-Water/Observation-Wells/Vanscoy/) and the website has clear description about pumping from 2003 to 2007. The pumping impacts are not included in our model and we tend to study the impacts of climate to groundwater, therefore sites that have strong anthropogenic influences are removed from this study.

[Figure]

Fig. S3. An example of anthropogenic pumping on groundwater level in Vanscoy, SK. The pumping from 2003 to 2007 has a strong drawdown of water level about 13 m and the slow recovery takes almost 10 years returning to its normal level.

*3) Related to the culling criteria, we evaluated the Alberta groundwater observation well data in a comparison with GRACE (Huang et al. 2016, Hydrogeology Journal v24, 1663-1680). You might be able to use Table 1 from that paper to increase the number of observation records. Also, you might be able to incorporate the GRACE comparison to water level data into your discussion and conclusions.*

Thank you for your comment and this is a very useful suggestion. Whilst in Huang et al. 2016 the 36 sites are selected to evaluate the GRACE terrestrial water storage (TWS) anomaly rather than the water table depth (WTD) in this study. Therefore, the WTD anomaly or variation is the focus of Huang et al. 2016 and the records from these 36 sites have demonstrated a range of depth from shallow to deep, as well as from surficial and confined aquifer.

However, the groundwater model we used in this study is an unconfined shallow aquifer below 2-m soil layers, therefore we chose only to evaluate the wells with recorded WTD within 5 m below surface, which is also pointed out by other studies as the earth's critical zone where water table could have critical impacts to the land and above atmosphere. Thus, we cannot use several deep groundwater sites as in Huang's paper.

*Technical Corrections*

*L17: Typo "on groundwater recharge rates"*

Done.

*L21: Is "mismatch" really the correct term? What you're describing is a parameter that is not represented in the model adequately. The resultant water table depth is mismatched, but the parameter is misrepresented.*

Done. We remove this sentence from the abstract as there are multiple reasons of WTD mismatch. Thanks for the correction of "misrepresented parameters".

*L23: Type "delaying the time…"*

Done.

*L47: A reference for the general concept of recharge and frozen soil would be useful (e.g. Hayashi)*

Reference added (Niu and Yang 2006; Mohammed et al., 2018).

*L61: Typo "discharge to rivers"*

Done, thanks for the correction.

*L64-65: Suggested edit "…snowmelt recharge to reach the water table, the previously upward water movement by capillary effect to reverse and move downwards, and allow the water table to rise to…"*

Done. Thank you for the suggestion.

*L66: Suggest removing "and desiccates the soil", as this starts to invoke ideas of seasonally varying parameters.*

Done.

*L76: Provide a reference for the 5-40mm/yr example.*

Thanks. Reference (Hayashi et al., 2016) is added and the paragraph is moved to discussion.

*L80: Typo "…this is challenging to represent in current…"*

Done, thank you for the correction.

*L85: Typo "suggested"*

Done.

*L97: Typo "groundwater models"*

Done.

*L128: What is meant by "groundwater evolution"? Do you simply mean water table dynamics?*

Yes, thanks for the correction.

*L143: Typo "…from the WRF…"*

Thanks for the correction.

*L148: You might want to mention that most of the observation well data will be biased toward more permeable deposits (e.g. sand and gravel). Typically, provincial and state agencies don't monitor low permeability formation.*

Thank you for this information. Very helpful to include this in this paper.

*L153: Alberta Environment and Parks*

Done.

*L164: Provide an example timeframe*

Fig. S1b provides the observation availability of the groundwater wells within the 13-year simulation period.

*L165: How was anthropogenic effect determined?*

The anthropogenic effects were determined by the site description on the Saskatchewan Water Security Agency websites. An example of anthropogenic pumping is provided in Fig. S2.

*L259: Add "in the PPR" to the end of this sentence*
Done.
*L285-288: The model initialization process is unclear. Spin up times of 500 years and 4 years are mentioned here. Is the 4 yr period simply to account for grid size difference, and essentially following the 500 yr spin up in the previous model?*
Thank you for this quesiton. The 500-year spin-up uses a 30-year climatology recharge as upper boundary condition. And the 4-year spin-up uses the forcing from 2000 Oct to 2001 Sep, and runs this year continuesouly for 4 loops, accounting for grid size difference and a more realistic initial condition at the beginning of the simulation. In this revision, we have the opportunity to do a longer spin-up for 10-year loop.
*L317-318: Relation to Amazon with is not relevant and totally looks like self-citation here. The concept of infiltration response is pretty basic.*
Thank you for this comment. The Amazon study reference (Miguez-Macho et al., 2012) is a study applying the same groundwater model in Amazon rainforest, in which similar shortcomings are reported as in our study. Thus, we believe this is a relevant study of modeling water table depth using MMF groundwater scheme.
*L321: By "out-of-the-box" do you simply mean "uncalibrated"?*
Yes.
*L323: Instead of "further study" do you mean "preliminary study", because later in the manuscript modified parameters are used (i.e. REP) Section 3.2 and 3.3: A little bit of set-up is needed here. Are the results presented averages over a certain period? What timeframe for the water budget components correspond to? (3 months)*
Thanks for the quesiton. "Further study" refers to the later-on analysis of groundwater fluxes and water balance in section 3.2 and 3.3. The results presented averages over a monthly interval.
*L425: Typo ". . .in the locations of the observations well."*
Done.
*L448-453: This section kinda teases the reader. How do the future scenario results look with the REP parameters?*
Thanks for the question. We appreciate this question and please see Answer 1 for detailed response.
*L520: Typo "As a result. . ."*
Thanks for the correction.

Thank you for the comment. This comment is very good and contains several questions. We would break this down and answer it separately.

The order of the paragraphs has been reorganized in the manuscript. The objectives of this study have been moved ahead in Introduction. The prairie pothole wetlands and groundwater-wetland exchanges is now moved to Discussion. More details for the methodological set-up, such as frozen soil treatments in LSMs, have been added in Data and Methods.

We move L70-83 to the Discussion section. This paragraph in the original manuscript introduced important ecosystem services provided by prairie pothole wetlands and the interactions between groundwater flow and these wetlands. Fine-scale processes (usually from 10 to 100 m), such as snow drift, runoff fill-spill and groundwater recharge/discharge to wetlands, have complicated the water balance in these wetlands and are challenging to LSMs (usually from 1 to 100 km) (Hayashi et al). In this work, we configured our model domain at 4-km resolution, and we acknowledged that it is insufficient to resolve these fine-scale processes. This is a current limit and future challenge in this study area. We are currently developing sub-grid parameterization scheme to represent the presence of these fine-scale prairie pothole wetlands to the hydrological cycle, including ET and sub-surface flows. Please see the Discussion section for more details.

We add a paragraph in Introduction to provide background of frozen soil parameterizations in most common LSMs, such as CLM, NoahV3, and Noah-MP. Additionally, a description of Noah-MP frozen soil parameterizations is included in the Methods section. There are two options in Noah-MP for frozen soil permeability; option 1 is the default option in Noah-MP LSM and is adapted from Niu and Yang (2006); option 2 is inherited the Koren et al. (1999) scheme from NoahV3. We used the option 1 in our simulation. The option 1 assumes that a model grid cell consists of permeable and impermeable areas and thus uses the total soil moisture to compute hydraulic properties of the soil.  The option 2 uses only the liquid water volume to calculate hydraulic properties. Additionally, option 1 assumes that soil ice has a linear (smaller) effect on infiltration, generally simulates more permeable frozen soil than option 2, which assumes soil ice has a non-linear (greater) effect on soil permeability (Niu et al., 2011).   For this reason, the option 1 allows the soil water to move and redistribute more easily within the frozen soil.

*2. More information is needed on the criteria for selecting wells, such as required length of record and how anthropogenic effects were minimized. Further, comparing a coarse land surface model covering a large total area (the model area is not reported) to only 11 wells is concerning. From Fig. 1 it appears that quite a large portion of the PPR does not have any well coverage. I think it would be worth revisiting the criteria and including a few more wells out of the 160. If that's not possible, then more discussion regarding potential uncertainty in capturing groundwater dynamics in PPR sub-regions without wells (e.g. the southwest portion) is warranted.*

Thank you for this comment and your concern about selecting observation wells. We have revisited the groundwater well observations and our selecting criteria in this revision. We use the daily water table depth records from total 160 groundwater wells in the domain, including 72 from the USGS, 43 from Alberta Environment and Parks, and 45 from Saskatchewan Water Security Agency. The locations of these 446 wells are shown in Fig. S1, together with the mean WTD and the availability of the observational records within the simulation period, respectively. (The model domain contains 401 x 396 grid points and is added in the Methods section).

We revisited the criteria to select these groundwater wells: (1) the location of the well is close to the PPR region; (2) a sufficiently long record during the simulation period. We define the observation availability as the available observation period within the 13-year simulation period and select wells with observation availability greater than 80%; (3) unconfined aquifer with shallow groundwater (mean WTD > -5 m); and (4) has little anthropogenic influence.

After these culling processes, 33 wells are selected, including 6 from Alberta, 13 from Saskatchewan, and 14 from the U.S., and the locations of these 33 wells are shown in Fig. S1c and their information in Table S1. Table S2 also provides the statistics, including mean and standard deviation of WTD, for these 33 sites, from observation and our groundwater model. The complete timeseries of these 33 sites are shown at the end of this response.

[Figure]

Fig. S1. The locations of the 160 groundwater wells in the PPR region and their (a) mean WTD values; (b) observation record availability; (c) the locations of 33 groundwater wells that have shallow groundwater level and long observation record (> 80%). A complete list of their information is presented in Table S1.

Table S1. Information about the selected 33 wells in the Prairie Pothole Region.

| Site Name/ Site No. | Lat | Lon | Elevation | Aquifer type | Aquifer Lithology | Model Elevation | Model Soil type |
|---|---|---|---|---|---|---|---|
| Devon 0162 | 53.41 | -113.76 | 700.0 | Unconfined | Sand | 697.366 | Sandy loam |
| Hardisty 0143 | 52.67 | -111.31 | 622.0 | Unconfined | Gravel | 633.079 | Loam |
| Kirkpatrick Lake 0229 | 51.95 | -111.44 | 744.5 | Semi-confined | Sandstone | 778.311 | Sandy loam |
| Metiskow 0267 | 52.42 | -110.60 | 677.5 | Unconfined | Sand | 679.516 | Loamy sand |
| Wagner 0172 | 53.56 | -113.82 | 670.0 | Surficial | Sand | 670.845 | Silt loam |
| Narrow Lake 252 | 54.60 | -113.63 | 640.0 | Unconfined | Sand | 701.0 | Clay loam |
| Baildon 060 | 50.25 | -105.50 | 590.184 | Surficial | - | 580.890 | Sandy loam |
| Beauval | 55.11 | -107.74 | 434.3 | Intertill | Sand | 446.5 | Sandy loam |
| Blucher | 52.03 | -106.20 | 521.061 | Intertill | Sand/Gravel | 523.217 | Loam |
| Crater Lake | 50.95 | -102.46 | 524.158 | Intertill | Sand/Gravel/Clay | 522.767 | Loam |
| Duck Lake | 52.92 | -106.23 | 502.920 | Surficial | Sand | 501.729 | Loamy sand |
| Forget | 49.70 | -102.85 | 606.552 | Surficial | Sand | 605.915 | Sandy loam |
| Garden Head | 49.74 | -108.52 | 899.160 | Bedrock | Sand/Till | 894.357 | Clay loam |
| Nokomis | 51.51 | -105.06 | 516.267 | Bedrock | Sand | 511.767 | Clay loam |
| Shaunavon | 49.69 | -108.50 | 896.040 | Bedrock | Sand/Till | 900.433 | Clay loam |
| Simpson 13 | 51.45 | -105.18 | 496.620 | Surficial | Sand | 493.313 | Sandy loam |
| Simpson 14 | 51.457 | -105.19 | 496.600 | Surficial | Sand | 493.313 | Sandy loam |
| Yorkton 517 | 51.17 | -102.50 | 513.643 | Surficial | Sand/Gravel | 511.181 | Loam |
| Agrium 43 | 52.03 | -107.01 | 500.229 | Intertill | Sand | 510.771 | Loam |
| 460120097591803 | 46.02 | -97.98 | 401.177 | Alluvial | Sand/Gravel | 400.381 | Sandy loam |
| 461838097553402 | 46.31 | -97.92 | 401.168 | - | Sand/Gravel | 404.719 | Clay loam |
| 462400097552502 | 46.39 | -97.92 | 409.73 | - | Sand/Gravel | 407.405 | Sandy loam |
| 462633097163402 | 46.44 | -97.27 | 325.52 | Alluvial | Sand/Gravel | 323.728 | Sandy loam |
| 463422097115602 | 46.57 | -97.19 | 320.40 | Alluvial | Sand/Gravel | 314.167 | Sandy loam |
| 464540100222101 | 46.76 | -100.37 | 524.91 | - | Sand/Gravel | 522.600 | Clay loam |
| 473841096153101 | 47.64 | -96.25 | 351.77 | Surficial | Sand/Gravel | 344.180 | Loamy sand |
| 473945096202402 | 47.66 | -96.34 | 327.78 | Surficial | Sand/Gravel | 328.129 | Sandy loam |
| 474135096203001 | 47.69 | -96.34 | 325.97 | Surficial | Sand/Gravel | 327.764 | Sandy loam |
| 474436096140801 | 47.74 | -96.23 | 341.90 | Surficial | Sand/Gravel | 336.210 | Sandy loam |
| 475224098443202 | 47.87 | -98.74 | 451.33 | - | Sand/Gravel | 450.463 | Sandy loam |
| 481841097490301 | 48.31 | -97.81 | 355.61 | - | Sand/Gravel | 359.568 | Clay loam |
| 482212099475801 | 48.37 | -99.79 | 488.65 | - | Sand/Gravel | 488.022 | Sandy loam |
| CRN Well WLN03 | 45.98 | -95.20 | 410.7 | Surficial | Sand/Gravel | 411.4 | Sandy loam |

Table S2. Statistics of mean and standard deviation of WTD for the selected 33 wells in the Prairie Pothole Region. Bold number indicates the REP run has improved results than the CTRL run.

| Site Name/Number | OBS_mean | CTRL_mean | REP_mean | OBS_std | CTRL_std | REP std |
|---|---|---|---|---|---|---|
| Devon 0162 | -2.46 | -2.69 | **-2.38** | 0.43 | 0.45 | 0.09 |
| Hardisty 0143 | -2.44 | -8.91 | **-6.88** | 0.41 | 0.64 | **0.36** |
| Kirkpatrick Lake 0229 | -4.22 | -4.03 | -3.45 | 0.43 | 0.98 | **0.22** |
| Metiskow 0267 | -2.54 | -5.39 | **-4.43** | 0.34 | 0.78 | **0.55** |
| Narrow Lake 252 | -2.31 | -4.81 | **-3.75** | 0.28 | 0.60 | 0.51 |
| Wagner 0172 | -2.14 | -8.06 | **-2.70** | 0.48 | 0.37 | 0.21 |
| Baildon 060 | -2.80 | -3.29 | **-3.20** | 0.47 | 0.58 | 0.30 |
| Beauval | -3.78 | -4.85 | **-4.20** | 0.44 | 0.56 | 0.32 |
| Blucher | -2.20 | -4.24 | **-2.16** | 0.3 | 0.92 | **0.26** |
| Crater Lake | -4.33 | -3.97 | -3.64 | 1.1 | 0.4 | 0.28 |
| Duck Lake | -3.65 | -3.69 | -3.17 | 0.54 | 0.41 | **0.62** |
| Forget | -2.28 | -2.37 | **-2.23** | 0.33 | 0.17 | 0.19 |
| Garden Head | -3.67 | -4.85 | **-3.77** | 0.88 | 0.70 | 0.30 |
| Nokomis | -1.04 | -2.70 | **-2.17** | 0.23 | 0.55 | **0.17** |
| Shaunavon | -1.62 | -4.41 | **-2.58** | 0.42 | 0.69 | 0.20 |
| Simpson 13 | -4.82 | -4.83 | -3.02 | 0.31 | 0.91 | **0.17** |
| Simpson 14 | -2.03 | -2.61 | **-1.82** | 0.34 | 0.18 | **0.27** |
| Yorkton 517 | -2.87 | -3.97 | **-1.98** | 0.8 | 0.46 | 0.32 |
| Agrium 43 | -2.66 | -3.75 | **-3.38** | 0.32 | 1.05 | **0.36** |
| 460120097591803 | -1.44 | -2.33 | **-1.63** | 0.56 | 0.24 | **0.50** |
| 461838097553402 | -1.17 | -2.32 | **-1.68** | 0.27 | 0.24 | 0.43 |
| 462400097552502 | -4.9 | -5.61 | **-5.37** | 0.29 | 0.09 | **0.17** |
| 462633097163402 | -1.18 | -1.49 | **-1.02** | 0.46 | 0.29 | **0.54** |
| 463422097115602 | -1.36 | -2.28 | **-1.66** | 0.34 | 0.23 | 0.49 |
| 464540100222101 | -2.02 | -3.64 | **-2.78** | 0.52 | 0.43 | 0.32 |
| 473841096153101 | -0.77 | -1.48 | **-1.37** | 0.24 | 0.18 | 0.51 |
| 473945096202402 | -1.59 | -1.58 | -1.56 | 0.32 | 0.24 | 0.51 |
| 474135096203001 | -0.72 | -1.48 | **-1.30** | 0.33 | 0.25 | 0.54 |
| 474436096140801 | -2.44 | -2.29 | -1.96 | 0.39 | 0.21 | **0.40** |
| 475224098443202 | -4.52 | -4.28 | -5.31 | 0.75 | 0.52 | 0.34 |
| 481841097490301 | -4.39 | -4.24 | -4.58 | 0.79 | 0.28 | 0.17 |
| 482212099475801 | -2.13 | -2.32 | **-2.26** | 0.24 | 0.20 | 0.17 |
| CRN WLN 03 | -2.04 | -2.18 | -1.88 | 0.24 | 0.18 | 0.43 |

*3. More information is needed on the climate change scenarios such as what emissions scenario was used and what sort of temperature increase does that roughly translate to?*

Thank you for this comment, we have added the information about emission scenario for the PGW forcing. The climate change forcing used in this LSM study is from a regional convective-permitting modeling project in North America, called WRF CONUS. Liu et al. (2017) discussed the WRF CONUS project in details. The CTRL run is forced with 6-hour ERA-Interim data, representing the current climate. The PGW run is forced with the ERA-Interim data plus a climate perturbation derived from CMIP5 ensemble under the RCP8.5 emission scenario, representing the future climate change till the end of 21$^{st}$ century. Fig. 4 and 5 in the manuscript show the temperature and precipitation change in PGW-CTRL. The most significant warming occurs in the winter over northern region and mountainous region in Alberta, warming up to 8 °C. An overall increase in precipitation is shown in Fig. 5, except in summer in the southeast, about 50 to 100 mm reduction.

*4. If the REP model performed better against observational data, then why didn't the authors choose to run the climate change scenario using that parameterization? Including such a simulation would give the reader a sense of how sensitive projections are to model parameterization. If including this simulation is too computationally expensive, then at least some discussion of how the climate projections might be sensitive to.*

We appreciate that both reviewers have asked a question about the responses of REP soil under future PGW climate forcing (PGW forcing). This is also an important point we need better elaborate in the manuscript and in this reply.

We revisit the observational groundwater wells and select 33 out of 160 wells (see Answer 2) and replace the default soil with sand in these 33 locations. For the rest of the domain, we keep the default soil type from the 1-km global soil map. The complete list of 33 groundwater observation wells and the modeled WTD with default (DEF, blue lines) soil and REP soil (red lines) are in Fig. S3 at the end of this document. We also conducted a simulation with REP soil under PGW climate. Ten sites are presented here as they show diverse results in these sites (see Fig. S2).

In general, under PGW climate, WTD rises due to increased precipitation and recharge. For some sites, the rise of WTD is more obvious in DEF soil rather than REP soil (e.g. Kirkpatrick Lake, Hardisty, Metiskow and 48184097490301). This is because the WTD under CTRL_REP is already higher than the WTD in CTRL_DEF and the $Q_r$ term (groundwater discharge to rivers) is parameterized as the gradient between WTD and riverbed (Eq. (8)). As a loss term in the groundwater flux, $Q_r$ is stronger in REP soil than in DEF soil and the climate change impacts on WTD rise is less prominent in REP soil than in DEF soil. On the other hand, there are some sites where PGW has little impacts on WTD, such as Simpson, Duck Lake and 482212099475801.

On point scale, given these diverse results over a limited number of sites, it is difficult to draw a universal conclusion but keep in mind the uncertainties and sensitivity of modeled WTD to soil parameters. On regional scale, the modifications of soil type at these 33 sites have little contribution to the large domain (401 x 396 grid points). Thus, our results of regional averaged water budget analysis in eastern and western PPR (Fig. 8 & 9) still hold. An ideal method to address this is to obtain sufficient information on soil properties accounting for horizontal and vertical heterogeneity. This is an on-going project that we are working on with the support from the Global Water Futures project and future improvements can be expected.

[Figure]

Fig. S2, the WTD dynamics of the observation and 4 model simulations: the two blue lines for default soil type (DEF), and two red lines for REP soil type (changed from default to sand); and solid lines for current climate (CTRL) and dashed line for future climate (PGW).

*5. There is a fairly major typographical error in the text and figure captions. Delta S should be equal to R + Qlat – Qr (according to equation 4), but it is repeatedly written as R + Qlat + Qr (equation 10, for example) in the paper. This is hopefully only typographical, as that would result in large errors in reported changes in storage.*

Thank you very much for pointing out this typo mistake. The *Qr* term characterizes the groundwater discharge to maintain river flow and in the model this term is always positive, meaning water flow from aquifer to riverbed. This is a loss term to the groundwater aquifer and that's why there is a negative sign before the term. I have corrected this in the manuscript, please see.

*6. The figure captions in the text often do not match the figure captions associated with the figures. Further, the authors should write out fully descriptive figure captions, including defining acronyms, such that the figures would be able to be read on their own. There are currently several figure captions that simply says, "same as Fig. xx."*
Thank you for the comment. Figure captions have been changed.

*7. Finally, the timing and amount of thaw is a key control on recharge projections, but the authors do not explore or discuss how well their model captures freeze-thaw dynamics at the regional scale. This is related to my earlier comment, that the authors need to explain how this process is represented in their model. Some discussion of how future studies could improve upon this methodology to capture this important and heterogeneous process would also add strength to the paper.*

Thank you very much and we appreciate this comment. In this revision, we introduced the frozen soil parameterizations in Noah-MP and other LSMs in the Introduction and Methods section. Although it is still a challenge to explore freeze-thaw dynamics on regional scale, we include some discussion on this matter and hope it can encourage future studies.

To our knowledge, there is no direct observation of soil ice for large region coverage. Most of the existing soil ice measurement are on local scale, for example, measurement from the FLUXNET sites, and have been used in for model evaluation (Niu and Yang 2006; Niu et al., 2011). Yang et al. (2011) provided a regional analysis on runoff, using the University of New Hampshire-Global Runoff Data Center dataset, and inferred the improved runoff simulation is the more permeable frozen soil in Noah-MP. These contents are also added to the Discussion as well.

*Specific comments:*
*There were quite a number of typographical errors, a few of which I will list here, but I do recommend the authors go back over the manuscript with a closer eye for spelling and grammatical errors.*

*Line 55: use a different acronym for precipitation- PR is too close to PPR*
*Lines 64-65: rewrite for grammar*
Thanks for the correction, done.

*Line 76: provide citation for recharge estimate*
Thanks for the correction, reference (Hayashi et al., 2016) added.

*Line 77: rewrite for grammar*
*Line 80: rewrite for grammar*
Done. This paragraph is now moved to Discussion.

*Lines 93-94: studies of regional climate change impacts to hydrology in N. America:*
*Niraula et al., 2017a,b, Christensen et al., 2004*
Thanks for these citations, added in the reference list.

*Line 148: Observational data*
*Line 181: define offline mode*
*Lines 341 and 350: under current/future climate conditions*
Thank you for the correction.

*Line 372: what is Qdrain?*
Should be "negative recharge". Corrected in the manuscript.

*Line 474: include citation*
Thank you for the reminder. Reference (Pokhrel et al., 2014) added in the list.

*Lines 520-522: rewrite for grammar*
Done.

*Table 1: either use descriptive column names or define any abbreviation used. Add units where needed.*
Thank you, definitions of abbreviation are added.

**Supplemental Materials - WTD dynamics from 33 groundwater wells in the PPR**
Alberta Environment and Parks

[Figure]

Saskatchewan Water Securtiy Agency

[Figure]

[Figure]

Simpson 14

[Figure]

Yorkton 517

[Figure]

Agrium 43

USGS

460120097591803

461838097553402

462400097552502

462633097163402

463422097115602

464540100222101

473841096153101

473945096202402

474135096203001

474436096140801

[Figure]

Fig. S4. WTD dynamics from observational wells and CTRL model with default soil (DEF, blue lines) and replacing default soil with sandy soil (REP, red lines) for the 33 sites in the PPR.

[revised manuscript text omitted]

Deleted: [1] Groundwater (GW) is an important source of freshwater for human beings. The domestic needs of about half of the world's population (UNESCO, 2004) and 38% of the global water demand for irrigation are provided by groundwater (Siebert et al., 2010). In the Canadian prairies, more than 30% of the population relied on groundwater in 1996 (Statistics Canada, 1996). In a more recent survey, while 90% of the municipal population is now provided by surface water sources, more than 50% of the population living in rural areas are still relying on groundwater sources in Canada (Environment Canada, 2011). In the U.S., up to 90% of water for drinking and irrigation are provided by groundwater across different parts of the country (National Research Council, 2003).¶
¶
[2] The groundwater flows in cold regions exhibit unique hydrological characteristics due to the hydraulic isolation, induced by seasonally frozen soil (Ireson et al., 2013). As frozen soils reduce permeability and snow accumulates during winter, the timing of groundwater recharge is controlled by the snowmelt and soil thaw period in spring. Previous work by Kelln et al. (2007) found that the timing of the recharge was associated strongly with soil thaw, rather than snowmelt, and occurred one to six weeks later than snowmelt. On the other hand, previous observations in a glacial-till site, where groundwater flow to underlying aquifer and lateral flow are small, have suggested the decline of water table during winter is related to an upward water transport to the freezing front (Remenda et al., 1996). ¶
¶
[3]

[revised manuscript text omitted]

+ Numbering Style: a, b, c, … + Start at: 1 + Alignment:
Left + Aligned at:  0.11 cm + Indent at:  0.74 cm

[Figure]

**Fig. 2** Structure of the Noah-MP LSM coupled with MMF groundwater scheme, the top 2-m soil of 4 layers whose thicknesses
are 0.1, 0.3, 0.6 and 1.0 m. An unconfined aquifer is added below the 2-m boundary, including an auxiliary layer and the saturated
aquifer. Positive flux of $R$ denotes downward transport. Two water table are shown, one within the 2-m soil and one below,
indicating that the model is capable to deal with both shallow and deep water table.
[Figure]

**Deleted: Fig. 2** Structure of the Noah-MP LSM coupled with MMF groundwater scheme, the top 2-m soil is consist of 4 layers whose depth are 0.1, 0.3, 0.6 and 1.0 m. An unconfined aquifer is added below the 2-m boundary, including an auxiliary layer and the saturated aquifer. Positive flux of $R$ denotes downward transport. Two water table are shown, one within the 2-m soil and one below, indicating that the model is capable to deal with shallow as well as deep water table. ¶

[Figure]

**Fig. 3** Evaluation of the annual precipitation from two model products (b, f), WRF CONUS and NARR against
rain gauge observation (a, e), their bias (c, g) and percentage bias (d, h).

[Figure]

**Fig. 4** Seasonal Accumulated precipitation from current climate (CTRL, top), future climate (PGW, middle) and
projected change (PGW-CTRL, bottom) in forcing data.

[Figure]

**Fig. 5** Seasonal temperatures from current climate (CTRL, top), future climate (PGW, middle) and projected
change (PGW-CTRL, bottom) in forcing data.

[Figure]

[Figure]

**Fig. 6.** WTD (m) bias from CTRL simulation and timeseries from 8 groundwater wells in PPR. See Table 2 CTRL
column for the model statistics and supplemental materials for complete timeseries from 33 wells.

[Figure]

**Fig. 7** Seasonal accumulated total groundwater fluxes ($R+Q_{lat} - Q_r$) for current climate (CTRL, top), future
climate (PGW, middle) and projected change (PGW-CTRL, bottom) in forcing data. Black dashed lines in PGW-
CTRL separate the PPR into eastern and western halves.

[Figure]

```
storage change:      1.763 mm
recharge change:     4.152 mm
river flux change:   2.260 mm
lateral flux change: -0.001 mm
```

**Fig. 8** Water budget analysis in the eastern PPR in (a) CTRL, (b) PGW and (c) PGW – CTRL. Water budget terms
include: (1) *PR & ET*, (2) surface snow, surface runoff and underground runoff (*SNOW*, *SFCRUN*, and *UDGRUN*),
(3) change of soil moisture storage (soil water, soil ice and total soil moisture, $\Delta SMC$) and (4) groundwater fluxes
and the change of groundwater storage ($R$, $Q_{lat}$, $Q_r$, $\Delta S_g$). The annual mean soil moisture change (PGW-CTRL) is
shown with black dashed line in (3). The Residual term is defined as $Res = (R+Q_{lat}-Q_r)-\Delta S_g$ in (4). Note that in (a)
and (b) the accumulated fluxes and change in storage are shown in lines, whereas in (c) the difference in (PGW-CTRL)
is shown for each individual month in bars.

[Figure]

storage change:  5.390 mm
recharge change:  10.727 mm
river flux change:  5.207 mm
lateral flux change:  0.000 mm

**Fig. 9** Same as Fig. 8, but for the **western PPR.**

[Figure]

[Figure]

**Fig. 10** Same as Fig. 6, the timeseries of simulated WTD from both default model (blue) and replacing soil type simulation, REP (red), REP is the additional simulation by replacing the default soil type in the model with sandy soil type.

| Page 3: [1] Deleted | Zhang, Zhe | 8/8/19 9:42:00 AM |
| Page 34: [2] Deleted | Zhang, Zhe | 8/20/19 2:16:00 PM |
| Page 34: [3] Deleted | Zhang, Zhe | 8/20/19 2:16:00 PM |
| Page 35: [4] Deleted | Zhang, Zhe | 8/20/19 2:17:00 PM |

---

## Referee Report (RR1)

Modeling groundwater responses to climate change in the Prairie Pothole Region

Reviewed by Katie Markovich

*Zhang et al.* use a land surface model coupled to a two-way groundwater dynamics model to explore the response of groundwater to climate change in the Prairie Pothole Region (PPR) of North America. The study is worthwhile due to the need to explore the hydrologic response to climate change at the regional scale in the PPR.

The authors sufficiently addressed my technical concerns from the first round of revisions. However, some minor issues remain in the presentation of their study.

Minor comments:

Line 51: what is "above soils" ?

Line 59: It would be better to actually review these studies included in the citation, as they support the idea that regional-scale simulation is necessary, and they have contributed important results to that end.

Lines 71-75: ParFlow-CLM simulates three-dimensional flow in the unsaturated and saturated zone, two-dimension flow on the surface, and a two-way exchange between the surface and subsurface. Thus, I am not sure where the conclusion from Line 75 is coming from.

Line 85: induce

Line 137: Here you say 32 wells but Figure 1 says 33.

Line 141: formations

Line 288: predicts a deep bias. (How deep is this bias?)

Line 291: the water table

Line 308: These hydrogeological

Line 352: the current and future climate

Line 356: rainfall events

Figures 9&10: The authors claim in their response to have fixed the figure captions, but these still have captions that read "same as [previous figure]." These need to be rewritten to be descriptive and standalone.

Figures 6&10: A color legend needs to be included on the figure for black, blue, and red lines. In addition a label containing units for the central map color bar needs to be added.

---

## Author Response (AR2)

**Response to Reviewer#2 and Editor**

(hess-2019-155)

Zhe Zhang, Yanping Li, Michael Barlage, Fei Chen, Gonzalo Miguez-Macho, Andrew Ireson, and Zhenhua Li

We appreciate the editor and two reviewers. They have put into time and effort to help us improve this article and their comments and suggestions are supportive and helpful. In the following text, the general response will be in red, original reviewers' comments and questions in *italic style* and our responses in blue texts.

**Responses to Reviewer#2:**

*Zhang et al. use a land surface model coupled to a two-way groundwater dynamics model to explore the response of groundwater to climate change in the Prairie Pothole Region (PPR) of North America. The study is worthwhile due to the need to explore the hydrologic response to climate change at the regional scale in the PPR.*
*The authors sufficiently addressed my technical concerns from the first round of revisions. However, some minor issues remain in the presentation of their study.*

*Line 51: what is "above soils" ?*
Thank you for the question. In this context, the "above soils" was referred to the unsaturated soils which is above the saturated aquifer. We have changed to "subsurface soils" in the manuscript revision2, see L50-51.

*Line 59: It would be better to actually review these studies included in the citation, as they support the idea that regional-scale simulation is necessary, and they have contributed important results to that end.*
We really appreciate this idea and it is a good idea to review these cited papers, see L59-65. These papers are good examples that climate change and groundwater modeling studies are necessary because they are important for water management decision making. These cited papers also identify the limitation of previous studies in this area, such as poor groundwater representation in models and uncertainties in climate change projections.

*Lines 71-75: ParFlow-CLM simulates three-dimensional flow in the unsaturated and saturated zone, two- dimension flow on the surface, and a two-way exchange between the surface and subsurface. Thus, I am not sure where the conclusion from Line 75 is coming from.*
Thank you for correcting this confusion. In the cited paper (Maxwell and Miller 2005), CLM and ParFlow was coupled as a single-column model. We have changed the description. The more advanced features mentioned by the reviewer are reviewed in L78-81 and Maxwell et al. (2015) is cited.

*Line 85: induce*
Done

*Line 137: Here you say 32 wells, but Figure 1 says 33.*
Thank you. This should be 33.

*Line 141: formations*
Done.

*Line 288: predicts a deep bias. (How deep is this bias?)*
Deep bias about 5-m as in Fig. 6, see L297.

*Line 291: the water table*
Done.

*Line 308: These hydrogeological*
Done.

*Line 352: the current and future climate*
Done.

*Line 356: rainfall events*
Done.

*Figures 9&10: The authors claim in their response to have fixed the figure captions, but these still have captions that read "same as [previous figure]." These need to be rewritten to be descriptive and standalone.*
Done.

*Figures 6&10: A color legend needs to be included on the figure for black, blue, and red lines. In addition, a label containing units for the central map color bar needs to be added.*
Done. Thank you for the reminder. Legend and unit are added.

**Responses to Editor:**

*Editor's comments:*

*Dear authors,*

*Thank you for your efforts to address the reviewer comments. These were mostly satisfactory. However, one of the reviewers had some additional comments and had noted that some changes had not been made. I had also found that some of the earlier changes that were said to be made had in fact not been made. Please go through the earlier referee reports and your responses to ensure that this is properly done. In addition, I have some minor comments (below). Please go through the entire manuscript for grammar as well.*

*P2, L21: the study does not consider mountainous regions. Change this to the northwestern part of the study area or something similar.*
Done.

*P2, L23: replace "bring forward" with "advancing"*
Done.

*P3, L40: add "the" in front of "freezing front"*
Done.

*P3, L41: remove the s from springs*
Done.

*P3, L45: remove the s from summers and falls*
Done.

*P4, L62: the objective of this paper "are" to…*
Done.

*On P4, the objectives are stated, but then the introduction and literature review continues after this. I recommend this be move toward the end of the section, as the second last paragraph.*
Thank you for the recommendation. The objective paragraph has been moved to the second last paragraph in the introduction.

*P5, L85: remove the s from induces*
Done.

*P5, L90: macropores "that" exist*
Done.

*P6, L 101: conduct dynamical downscaling. Also in L111.*
Done. Thank you for the correction.

*P6, L113: as follows*
Done.

*P42, Fig. 6: a legend in the figure to show which line is simulation and which is observation would be helpful. Or else move the text from P16, L285–286 into the figure caption.*
Done. This is also mentioned by the reviewer#2. Thank you both for the correction, legend and unit are added.

*P18, L323: remove the s from rises*
Done.

*P18, L329: fluxes show strong…*
Done.

*P23, L412: we show that the model…*
Done.

*P24, L450-451: fix the grammar with this sentence*
Done.

*P26, L486: remove the s from components*
Done.

*P26, L488: is challenging to what? A word is missing.*
Done.

*P26, L490-491: please fix the grammar*
Done. Thank you for the kind reminder.

[revised manuscript text omitted]

Font: 12 pt

| Page 12: [1] Formatted | Zhang, Zhe | 11/22/19 3:33:00 PM |
|---|---|---|

Font: 12 pt

| Page 12: [1] Formatted | Zhang, Zhe | 11/22/19 3:33:00 PM |
|---|---|---|

Font: 12 pt

| Page 12: [1] Formatted | Zhang, Zhe | 11/22/19 3:33:00 PM |
|---|---|---|

Font: 12 pt

| Page 12: [1] Formatted | Zhang, Zhe | 11/22/19 3:33:00 PM |
|---|---|---|

Font: 12 pt

| Page 12: [1] Formatted | Zhang, Zhe | 11/22/19 3:33:00 PM |
|---|---|---|

Font: 12 pt

| Page 12: [2] Formatted | Zhang, Zhe | 11/22/19 3:33:00 PM |
|---|---|---|

Font: 12 pt

| Page 12: [2] Formatted | Zhang, Zhe | 11/22/19 3:33:00 PM |
|---|---|---|

Font: 12 pt

| Page 12: [2] Formatted | Zhang, Zhe | 11/22/19 3:33:00 PM |
|---|---|---|

Font: 12 pt

| Page 12: [2] Formatted | Zhang, Zhe | 11/22/19 3:33:00 PM |
|---|---|---|

Font: 12 pt

| Page 12: [2] Formatted | Zhang, Zhe | 11/22/19 3:33:00 PM |
|---|---|---|

Font: 12 pt

| Page 12: [2] Formatted | Zhang, Zhe | 11/22/19 3:33:00 PM |
|---|---|---|

Font: 12 pt

| Page 12: [2] Formatted | Zhang, Zhe | 11/22/19 3:33:00 PM |
|---|---|---|

Font: 12 pt

| Page 12: [2] Formatted | Zhang, Zhe | 11/22/19 3:33:00 PM |
|---|---|---|

Font: 12 pt

| Page 12: [2] Formatted | Zhang, Zhe | 11/22/19 3:33:00 PM |
|---|---|---|

Font: 12 pt

| Page 12: [2] Formatted | Zhang, Zhe | 11/22/19 3:33:00 PM |
|---|---|---|

Font: 12 pt

| Page 12: [2] Formatted | Zhang, Zhe | 11/22/19 3:33:00 PM |
|---|---|---|

Font: 12 pt

| Page 12: [2] Formatted | Zhang, Zhe | 11/22/19 3:33:00 PM |
|---|---|---|

Font: 12 pt

| Page 12: [2] Formatted | Zhang, Zhe | 11/22/19 3:33:00 PM |
|---|---|---|

Font: 12 pt

| Page 12: [2] Formatted | Zhang, Zhe | 11/22/19 3:33:00 PM |
|---|---|---|

Font: 12 pt

| Page 12: [2] Formatted | Zhang, Zhe | 11/22/19 3:33:00 PM |
|---|---|---|

Font: 12 pt

| Page 12: [3] Formatted | Zhang, Zhe | 11/22/19 3:33:00 PM |
|---|---|---|

Font: 12 pt

| Page 12: [3] Formatted | Zhang, Zhe | 11/22/19 3:33:00 PM |
|---|---|---|

Font: 12 pt

| Page 12: [3] Formatted | Zhang, Zhe | 11/22/19 3:33:00 PM |
|---|---|---|

Font: 12 pt

| Page 12: [3] Formatted | Zhang, Zhe | 11/22/19 3:33:00 PM |
|---|---|---|

Font: 12 pt

| Page 12: [3] Formatted | Zhang, Zhe | 11/22/19 3:33:00 PM |
|---|---|---|

Font: 12 pt

| Page 12: [3] Formatted | Zhang, Zhe | 11/22/19 3:33:00 PM |
|---|---|---|

Font: 12 pt

| Page 12: [3] Formatted | Zhang, Zhe | 11/22/19 3:33:00 PM |
|---|---|---|

Font: 12 pt

| Page 12: [3] Formatted | Zhang, Zhe | 11/22/19 3:33:00 PM |
|---|---|---|

Font: 12 pt

| Page 12: [3] Formatted | Zhang, Zhe | 11/22/19 3:33:00 PM |
|---|---|---|

Font: 12 pt

| Page 12: [3] Formatted | Zhang, Zhe | 11/22/19 3:33:00 PM |
|---|---|---|

Font: 12 pt

| Page 12: [3] Formatted | Zhang, Zhe | 11/22/19 3:33:00 PM |
|---|---|---|

Font: 12 pt

| Page 12: [3] Formatted | Zhang, Zhe | 11/22/19 3:33:00 PM |
|---|---|---|

Font: 12 pt

| Page 12: [4] Formatted | Zhang, Zhe | 11/22/19 3:33:00 PM |
|---|---|---|

Font: 12 pt

| Page 12: [4] Formatted | Zhang, Zhe | 11/22/19 3:33:00 PM |
|---|---|---|

Font: 12 pt

| Page 12: [4] Formatted | Zhang, Zhe | 11/22/19 3:33:00 PM |
|---|---|---|

Font: 12 pt

| Page 12: [4] Formatted | Zhang, Zhe | 11/22/19 3:33:00 PM |
|---|---|---|

Font: 12 pt

| Page 12: [4] Formatted | Zhang, Zhe | 11/22/19 3:33:00 PM |
|---|---|---|

Font: 12 pt

| Page 12: [4] Formatted | Zhang, Zhe | 11/22/19 3:33:00 PM |
|---|---|---|

Font: 12 pt

| Page 12: [4] Formatted | Zhang, Zhe | 11/22/19 3:33:00 PM |
|---|---|---|

Font: 12 pt

| Page 12: [4] Formatted | Zhang, Zhe | 11/22/19 3:33:00 PM |
|---|---|---|

Font: 12 pt

| Page 12: [4] Formatted | Zhang, Zhe | 11/22/19 3:33:00 PM |
|---|---|---|

Font: 12 pt

| Page 12: [4] Formatted | Zhang, Zhe | 11/22/19 3:33:00 PM |
|---|---|---|

Font: 12 pt

| Page 12: [4] Formatted | Zhang, Zhe | 11/22/19 3:33:00 PM |
|---|---|---|

Font: 12 pt

| Page 12: [4] Formatted | Zhang, Zhe | 11/22/19 3:33:00 PM |
|---|---|---|

Font: 12 pt

| Page 12: [4] Formatted | Zhang, Zhe | 11/22/19 3:33:00 PM |
|---|---|---|

Font: 12 pt

| Page 12: [4] Formatted | Zhang, Zhe | 11/22/19 3:33:00 PM |
|---|---|---|

Font: 12 pt

| Page 12: [5] Formatted | Zhang, Zhe | 11/22/19 3:33:00 PM |
|---|---|---|

Font: 12 pt

| Page 12: [5] Formatted | Zhang, Zhe | 11/22/19 3:33:00 PM |
|---|---|---|

Font: 12 pt

| Page 12: [6] Formatted | Zhang, Zhe | 11/22/19 3:33:00 PM |
|---|---|---|

Font: 12 pt

| Page 12: [6] Formatted | Zhang, Zhe | 11/22/19 3:33:00 PM |
|---|---|---|

Font: 12 pt

| Page 12: [7] Formatted | Zhang, Zhe | 11/22/19 3:33:00 PM |
|---|---|---|

Font: 12 pt

| Page 12: [7] Formatted | Zhang, Zhe | 11/22/19 3:33:00 PM |
|---|---|---|

Font: 12 pt

| Page 12: [7] Formatted | Zhang, Zhe | 11/22/19 3:33:00 PM |
|---|---|---|

Font: 12 pt

| Page 12: [7] Formatted | Zhang, Zhe | 11/22/19 3:33:00 PM |
|---|---|---|

Font: 12 pt

| Page 12: [7] Formatted | Zhang, Zhe | 11/22/19 3:33:00 PM |
|---|---|---|

Font: 12 pt

| Page 12: [7] Formatted | Zhang, Zhe | 11/22/19 3:33:00 PM |
|---|---|---|

Font: 12 pt

| Page 12: [7] Formatted | Zhang, Zhe | 11/22/19 3:33:00 PM |
|---|---|---|

Font: 12 pt

| Page 12: [7] Formatted | Zhang, Zhe | 11/22/19 3:33:00 PM |
|---|---|---|

Font: 12 pt

| Page 12: [7] Formatted | Zhang, Zhe | 11/22/19 3:33:00 PM |
|---|---|---|

Font: 12 pt

| Page 12: [7] Formatted | Zhang, Zhe | 11/22/19 3:33:00 PM |
|---|---|---|

Font: 12 pt

| Page 12: [7] Formatted | Zhang, Zhe | 11/22/19 3:33:00 PM |
|---|---|---|

Font: 12 pt

| Page 12: [7] Formatted | Zhang, Zhe | 11/22/19 3:33:00 PM |
|---|---|---|

Font: 12 pt

| Page 12: [7] Formatted | Zhang, Zhe | 11/22/19 3:33:00 PM |
|---|---|---|

Font: 12 pt

| Page 12: [7] Formatted | Zhang, Zhe | 11/22/19 3:33:00 PM |
|---|---|---|

Font: 12 pt

| Page 12: [7] Formatted | Zhang, Zhe | 11/22/19 3:33:00 PM |
|---|---|---|

Font: 12 pt

| Page 12: [7] Formatted | Zhang, Zhe | 11/22/19 3:33:00 PM |
|---|---|---|

Font: 12 pt

| Page 12: [7] Formatted | Zhang, Zhe | 11/22/19 3:33:00 PM |
|---|---|---|

Font: 12 pt

| Page 12: [7] Formatted | Zhang, Zhe | 11/22/19 3:33:00 PM |
|---|---|---|

Font: 12 pt

| Page 12: [7] Formatted | Zhang, Zhe | 11/22/19 3:33:00 PM |
|---|---|---|

Font: 12 pt

| Page 12: [7] Formatted | Zhang, Zhe | 11/22/19 3:33:00 PM |
|---|---|---|

Font: 12 pt

| Page 12: [7] Formatted | Zhang, Zhe | 11/22/19 3:33:00 PM |
|---|---|---|

Font: 12 pt

| Page 12: [7] Formatted | Zhang, Zhe | 11/22/19 3:33:00 PM |
|---|---|---|

Font: 12 pt

| Page 12: [7] Formatted | Zhang, Zhe | 11/22/19 3:33:00 PM |
|---|---|---|

Font: 12 pt

| Page 12: [8] Formatted | Zhang, Zhe | 11/22/19 3:33:00 PM |
|---|---|---|

Font: 12 pt

| Page 12: [8] Formatted | Zhang, Zhe | 11/22/19 3:33:00 PM |
|---|---|---|

Font: 12 pt

| Page 12: [9] Formatted | Zhang, Zhe | 11/22/19 3:33:00 PM |
|---|---|---|

Font: 12 pt

| Page 12: [9] Formatted | Zhang, Zhe | 11/22/19 3:33:00 PM |
|---|---|---|

Font: 12 pt

| Page 12: [9] Formatted | Zhang, Zhe | 11/22/19 3:33:00 PM |
|---|---|---|

Font: 12 pt

| Page 12: [9] Formatted | Zhang, Zhe | 11/22/19 3:33:00 PM |
|---|---|---|

Font: 12 pt

| Page 12: [10] Formatted | Zhang, Zhe | 11/22/19 3:33:00 PM |
|---|---|---|

Font: 12 pt

| Page 12: [10] Formatted | Zhang, Zhe | 11/22/19 3:33:00 PM |
|---|---|---|

Font: 12 pt

| Page 12: [11] Formatted | Zhang, Zhe | 11/22/19 3:33:00 PM |
|---|---|---|

Font: 12 pt

| Page 12: [11] Formatted | Zhang, Zhe | 11/22/19 3:33:00 PM |
|---|---|---|

Font: 12 pt

| Page 12: [12] Formatted | Zhang, Zhe | 11/22/19 3:33:00 PM |
|---|---|---|

Font: 12 pt

| Page 12: [12] Formatted | Zhang, Zhe | 11/22/19 3:33:00 PM |
|---|---|---|

Font: 12 pt

---

## Author Response (AR3)

**Response to Editor (Re3)**

(hess-2019-155)

Zhe Zhang, Yanping Li, Michael Barlage, Fei Chen, Gonzalo Miguez-Macho, Andrew Ireson, and Zhenhua Li

Thank you for taking this article into consideration and your efforts are much appreciated. Please see the reply to Editor's comment in the following texts:

*Comments to the Author:*
*Dear authors,*
*Thank you for your efforts to address the latest set of comments. The manuscript is now ready to be published. there is just one last minor issue. The term "subsurface soils" in the intro on L51-52 is still unclear. Why not call it what you mean, which is "unsaturated soils above the saturated aquifer"?*
*Thank you.*

Thank you for this comment and suggestion. We have changed the term "subsurface soils" into "unsaturated soils" through the manuscript. This indeed makes a clearer sense. Thank you very much. See L51-52 in the manuscript.